EMBO
Molecular Medicine

# Gene signature driving invasive mucinous adenocarcinoma of the lung

Minzhe Guo[1,2,†], Koichi Tomoshige[1,†], Michael Meister[3], Thomas Muley[3], Takuya Fukazawa[4], Tomoshi Tsuchiya[5], Rebekah Karns[6], Arne Warth[7], Iris M Fink-Baldauf[1], Takeshi Nagayasu[5], Yoshio Naomoto[4], Yan Xu[1], Marcus A Mall[8] & Yutaka Maeda[1,*] ID

## Abstract

Though invasive mucinous adenocarcinoma of the lung (IMA) is pathologically distinctive, the molecular mechanism driving IMA is not well understood, which hampers efforts to identify therapeutic targets. Here, by analyzing gene expression profiles of human and mouse IMA, we identified a Mucinous Lung Tumor Signature of 143 genes, which was unexpectedly enriched in mucin-producing gastrointestinal, pancreatic, and breast cancers. The signature genes included transcription factors *FOXA3, SPDEF, HNF4A,* mucins *MUC5AC, MUC5B, MUC3,* and an inhibitory immune checkpoint *VTCN1/B7-H4* (but not *PD-L1/B7-H1*). Importantly, induction of FOXA3 or SPDEF along with mutant KRAS in lung epithelium was sufficient to develop benign or malignant mucinous lung tumors, respectively, in transgenic mice. FOXA3 and SPDEF induced *MUC5AC* and *MUC5B,* while HNF4A induced *MUC3* in human mucinous lung cancer cells harboring a *KRAS* mutation. ChIP-seq combined with CRISPR/Cas9 determined that upstream enhancer regions of the mucin genes *MUC5AC* and *MUC5B,* which were bound by SPDEF, were required for the expression of the mucin genes. Here, we report the molecular signature and gene regulatory network driving mucinous lung tumors.

**Keywords** FOXA3; IMA; MUC5AC/5B; SPDEF; VTCN1

**Subject Categories** Cancer; Chromatin, Epigenetics, Genomics & Functional Genomics; Respiratory System

## Introduction

Lung cancer is the leading cause of cancer death. Lung cancer is pathologically classified into two main types, non-small cell lung cancer (NSCLC, the most common type) and small cell lung cancer (SCLC). Lung adenocarcinoma (LUAD) is the most frequent type of NSCLC followed by lung squamous cell carcinoma (SqCC) and large cell carcinoma (LCC). Invasive mucinous adenocarcinoma of the lung (IMA) comprises ~5–10% of LUAD (estimated approximately 5,000 deaths/year by IMA in the US) whose tumor cells display goblet cell morphology containing abundant intracytoplasmic mucin (Hata *et al*, 2010; Kunii *et al*, 2011; Travis *et al*, 2011). Recent RNA-seq analyses demonstrated that *KRAS* mutation was the most frequent genetic alteration seen in IMA (40–62%) followed by *NRG1* fusion (7–27%; Fernandez-Cuesta *et al*, 2014; Nakaoku *et al*, 2014). IMA expresses mucins such as MUC5AC and MUC5B while lacking the transcription factor NKX2-1 (also known as TTF-1; Travis *et al*, 2011) that normally suppresses those mucin genes (Maeda *et al*, 2012). Since the majority of IMA carries "undruggable" *KRAS* mutations, few therapeutics have been identified other than chemotherapy. Determining the molecular mechanisms driving IMA is required to identify novel therapeutic targets.

Previously, we and others demonstrated that *Kras* lung cancer mouse models with reduced expression of *Nkx2-1* (*Kras^{G12D}; Nkx2-1^{+/−}* or *Kras^{G12D}; Nkx2-1^{flox/flox}*) developed mucinous lung tumors mimicking human IMA (Maeda *et al*, 2012; Snyder *et al*, 2013). Independent analyses of gene expression profiles using cDNA microarrays identified 287 genes (*Kras^{G12D};Nkx2-1^{+/−}*; Maeda *et al*, 2012) and 1381 genes (*Kras^{G12D}; Nkx2-1^{flox/flox}*; Snyder *et al*, 2013) as genes induced in mucinous lung tumors in the mouse models.

1 Perinatal Institute, Divisions of Neonatology, Perinatal and Pulmonary Biology, Cincinnati Children's Hospital Medical Center and the University of Cincinnati College of Medicine, Cincinnati, OH, USA
2 Department of Electrical Engineering and Computing Systems, University of Cincinnati, Cincinnati, OH, USA
3 Translational Research Unit, Thoraxklinik at University Hospital Heidelberg, Translational Lung Research Center Heidelberg (TLRC), Member of the German Center for Lung Research (DZL), Heidelberg, Germany
4 Department of General Surgery, Kawasaki Medical School, Okayama, Japan
5 Department of Surgical Oncology, Nagasaki University Graduate School of Biomedical Sciences, Nagasaki, Japan
6 Division of Biomedical Informatics, Cincinnati Children's Hospital Medical Center and the University of Cincinnati College of Medicine, Cincinnati, OH, USA
7 Institute of Pathology, Translational Lung Research Center Heidelberg (TLRC), Member of the German Center for Lung Research (DZL), University of Heidelberg, Heidelberg, Germany
8 Department of Translational Pulmonology, Translational Lung Research Center Heidelberg (TLRC), Member of the German Center for Lung Research (DZL), University of Heidelberg, Heidelberg, Germany
*Corresponding author. Tel: +1 513 803 5066; Fax: +1 513 636 7868; E-mail: yutaka.maeda@cchmc.org
†These authors contributed equally to this work

However, it remains unknown which genes are indeed expressed in human IMA. In the present study, we analyzed gene expression profiles of human IMA using RNA-seq and determined genes commonly expressed in the mouse models and human IMA as a gene signature for IMA. The signature included potential therapeutic target genes such as the immune checkpoint *VTCN1/B7-H4*. The uniqueness of the signature was further assessed using RNA-seq data from 230 LUAD specimens from The Cancer Genome Atlas (TCGA; Cancer Genome Atlas Research Network, 2014) and 598 human cancer cell lines from 20 different organs/tissues, including NSCLC cell lines (Klijn *et al*, 2015). The analysis using the signature and human IMA specimens further revealed that pro-mucous transcription factors FOXA3, SPDEF, and HNF4A in addition to the anti-mucous transcription factor NKX2-1 differentially regulate the expression of mucins and IMA-related genes in human lung cancer cells *in vitro*. Importantly, induction of FOXA3 or SPDEF along with KRAS^(G12D) in lung epithelium was sufficient to develop mucinous lung tumors *in vivo*. ChIP-seq analysis determined that SPDEF directly bound to the non-coding loci of the mucin genes *MUC5AC* and *MUC5B*, two major mucins expressed in the lung. Deletion of the SPDEF-bound regions using CRISPR/Cas9 significantly reduced the expression of the two mucins, indicating that these non-coding regions are functionally indispensable in inducing the expression of mucins in IMA. Here, we report the novel gene signature and gene regulatory mechanisms for IMA.

## Results

### Mucinous Lung Tumor Signature is enriched in specific cancer types

In order to identify genes that are highly expressed in human IMA, we performed mRNA-seq analysis using RNAs extracted from six human IMA and adjacent normal lung tissues and determined genes differentially expressed in human IMA (Figs 1A and EV1, and Datasets EV1 and EV2). We compared the genes induced in human IMA (Dataset EV2) with the genes induced in mouse mucinous lung tumors (Mouse IMA; Maeda *et al*, 2012; Snyder *et al*, 2013; Dataset EV3) and identified 143 genes that were commonly expressed in both mouse and human IMA as a "Mucinous Lung Tumor Signature" (Figs 1B and EV2, red rectangles). In order to assess whether the signature indeed represents gene expression profiles of human IMA, we retrieved the RNA-seq data of nine human IMA and nine normal lung specimens (Dataset EV4) from TCGA (Cancer Genome Atlas Research Network,

2014) and performed Gene Set Enrichment Analysis (GSEA; Subramanian *et al*, 2005). The signature (141 genes out of the 143 genes since *MUC5AC* and *MUC3A/B* were not included in the TCGA datasets) was highly enriched in human IMA (Enrichment Score = 0.81; Fig 1C), confirming that the signature can be used to assess human IMA. Interestingly, the signature selectively clustered the TCGA-human IMA cases that harbored *KRAS* mutations, which separated the cases harboring other genetic alterations, including fusion genes (Fig 1D and E). These results suggest that this signature represents human IMA cases, especially the ones that harbor a *KRAS* mutation, which is the most frequent driver mutation in the human IMA.

Since mouse IMA is considered to be a lung tumor with gastric differentiation (Snyder *et al*, 2013), we sought to determine whether the Mucinous Lung Tumor Signature is enriched in human gastric cancers. We retrieved the RNA-seq data of 598 cancer cell lines, including gastric and other cancer cell lines (Dataset EV4; Klijn *et al*, 2015), assessed the expression of the signature genes in the groups of the cell lines (Fig 2A) and calculated the GSEA enrichment for the signature in different cancer types (Fig 2B). Consistent with a previous report (Snyder *et al*, 2013), the signature was highly enriched in human stomach and colorectal cancer cells (Enrichment Score = 0.79 and 0.81, respectively). The signature was also highly enriched in human LUAD (Enrichment Score = 0.76), suggesting that human IMA maintains a lung lineage though it is morphologically distinct from non-mucinous LUAD. Unexpectedly, the signature was also highly enriched in human pancreatic and breast cancers (Enrichment Score = 0.78 and 0.73, respectively). Similar to IMA, pancreatic and breast cancers produce mucins (Kufe, 2009). These results suggest that human IMA harbors a unique gene expression profile including not only gastric genes but also other genes expressed in mucin-producing cancers.

### An immune checkpoint VTCN1 but not PD-L1 is expressed in human IMA

Among the signature genes (Figs 1B and EV2, red rectangles), we identified an immune checkpoint *VTCN1* (also known as *B7-H4*). Recent successes on cancer immunotherapy targeting the immune checkpoint PD-1 and/or PD-L1 (also known as B7-H1 or CD274) by therapeutic antibodies indicate that activating the immune system is beneficial for treating NSCLC in patients whose lung tumors express PD-L1 (Herbst *et al*, 2014; Garon *et al*, 2015; Hirsch *et al*, 2016; Smyth *et al*, 2016). VTCN1 is also expected to be an antibody-mediated therapeutic target for breast cancer (Leong *et al*, 2015; Smyth *et al*, 2016).

**Figure 1.   Mucinous lung tumor gene signature for human invasive mucinous adenocarcinoma of the lung (IMA).**

A   Shown are differentially expressed genes in human IMA. RNA-seq was performed using six human IMA patient cases (case 1–6) along with adjacent normal lung tissues.

B   143 genes of the Mucinous Lung Tumor Signature (three red rectangles) are in common between genes expressed in mouse IMA (Maeda *et al*, 2012; Snyder *et al*, 2013) and genes induced in human IMA (see Dataset EV2).

C   Gene Set Enrichment Analysis (GSEA) shows that the Mucinous Lung Tumor Signature is significantly enriched in TCGA-human IMA specimens (nine cases) compared to TCGA-normal lung specimens (nine cases). 141 genes out of the 143 genes were used for the analysis since the TCGA LUAD data do not include *MUC5AC* and *MUC3A/B*. ES and *P* were the "Enrichment Score" and "Nominal *P*-value", respectively, generated by GSEA.

D   Five out of the nine cases of TCGA-human IMA patient cases were highly clustered in 230 TCGA LUAD cases, including IMA and non-IMA cases, based on the Mucinous Lung Tumor Signature of the 141 genes. Top panel, pathology (red indicates IMA cases), driver gene mutations (fusion or mutation), sex, smoking status, and tumor stage are shown in *y*-axis. *X*-axis indicates 230 patient cases from the TCGA LUAD datasets. Bottom panel, expression level of the 141 genes in the Mucinous Lung Tumor Signature is shown (*y*-axis, the 141 genes). *X*-axis is as described above.

E   The nine human IMA patient cases from the TCGA LUAD datasets expressed the 140 genes out of the 141 genes (*TNFSF18* was not expressed). The Mucinous Lung Tumor Signature of the 141 genes clustered the IMA cases harboring KRAS mutation differentially from those harboring fusions and wild-type KRAS.

                                                                                  

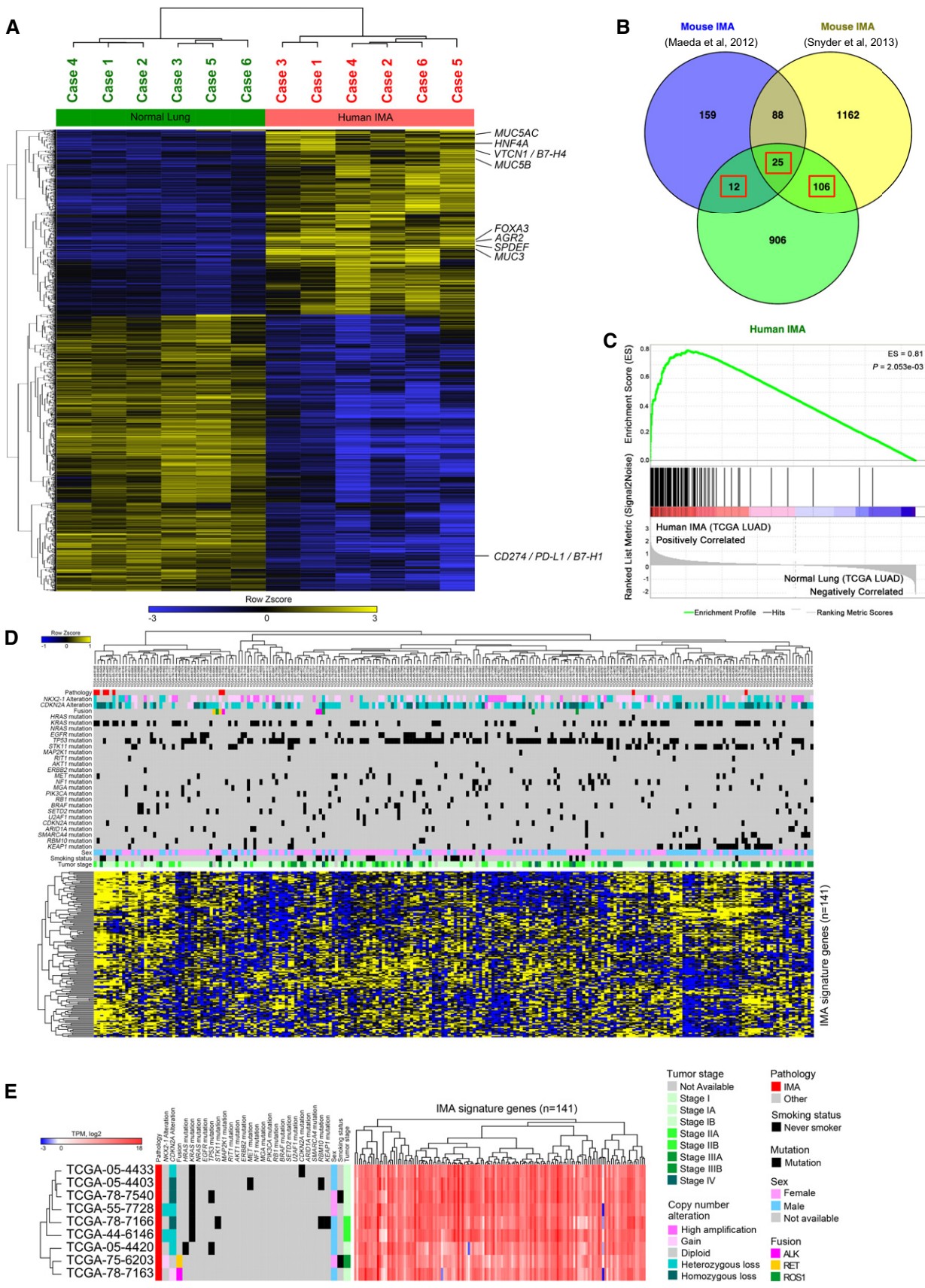

**Figure 1.**

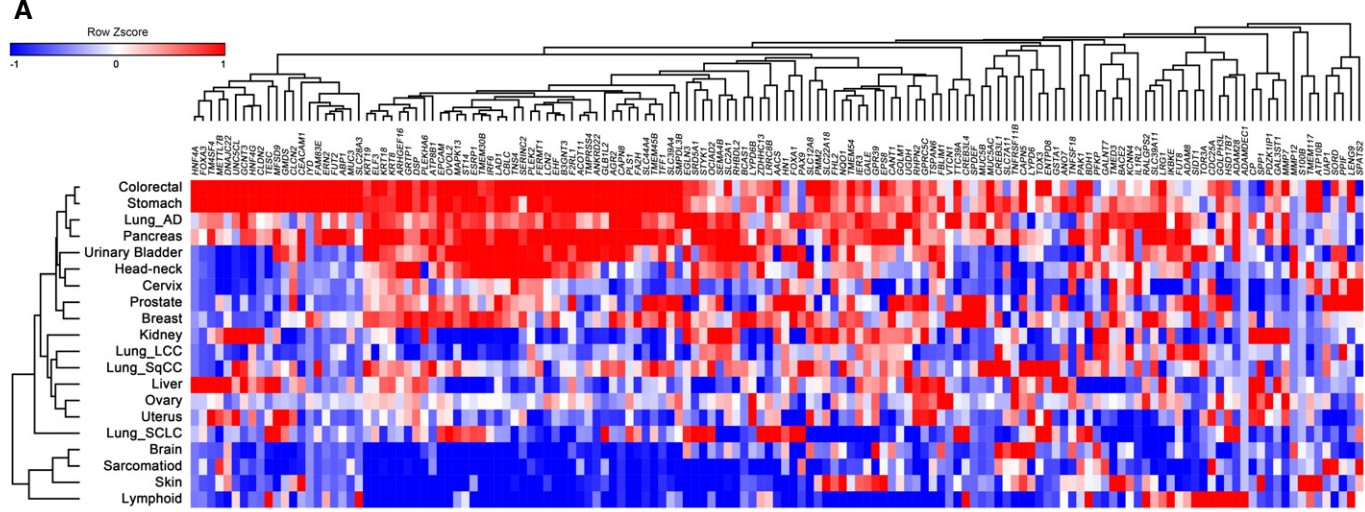

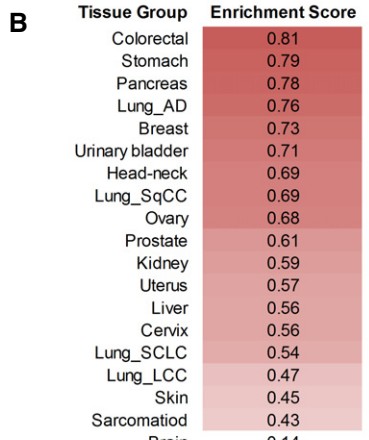

**Figure 2. The Mucinous Lung Tumor Signature is highly enriched in colorectal, stomach, pancreatic, lung (adenocarcinoma; LUAD), and breast cancers.**

A  Expression of the Mucinous Lung Tumor Signature genes (143 genes) in different cancers is shown. RNA-seq data from different cancers were obtained from multiple cancer cell lines (n = 598, Dataset EV4), which were reported previously (Klijn *et al*, 2015). Gene expression was measured in RPKM quantile-normalized and log2-transformed. The minimum of log2-transformed values was set to 0. The expression of a signature gene in a tissue group (cancer type) was measured as the average of the log2-transformed gene expression in the cell lines of the tissue group. Hierarchical clustering was performed using Pearson's correlation-based distance and average linkage.

B  Mucinous Lung Tumor Signature enrichment score for different cancers using Gene Set Enrichment Analysis (GSEA) is shown. Using the lymphoid cell lines as control, colorectal, stomach, lung AD, pancreas, breast, urinary bladder, head–neck, lung SqCC, and ovarian cancers were highly enriched with the Mucinous Lung Tumor Signature. Lung_AD: lung adenocarcinoma; Lung_LCC: lung large cell carcinoma; Lung_SqCC: lung squamous cell carcinoma; Lung_SCLC: small cell lung cancer.

Thus, we further investigated the expression of the immune checkpoints VTCN1 together with PD-L1 in human IMA. The expression of *VTCN1* mRNA was higher in human IMA than in normal control lung tissues; however, the expression of *PD-L1* mRNA was not induced in human IMA compared to normal control lung tissues (Figs 1A and EV1). The expression of VTCN1 protein was observed in 64% of human IMA, while the expression of PD-L1 protein was not observed in a majority of human IMA (< 10%; Fig 3A and B, and Dataset EV1). A binomial test indicates that the absence of PD-L1 but not that of VTCN1 is significant in human IMA (one-tailed binomial test: *P*-value = 1.794E-05). Next, we investigated the association of *VTCN1* or *PD-L1* with genes expressed in human IMA using two RNA-seq datasets from 105 NSCLC cell lines (Klijn *et al*, 2015) and 230 TCGA LUAD cases (Cancer Genome Atlas

Research Network, 2014). 38 genes out of the top 200 *VTCN1*-highly correlated genes in the 105 NSCLC cell lines were induced in human IMA, while only seven out of the top 200 *PD-L1*-highly correlated genes in the 105 NSCLC cell lines were induced in human IMA (Fig 3C, left panel and Dataset EV5). A similar association was observed using the 230 TCGA LUAD cases (Fig 3C, right panel and Dataset EV5), suggesting a gene signature associated with *VTCN1*, rather than with *PD-L1*, is related to human IMA. An unbiased clustering analysis using the RNA-seq dataset from the 105 NSCLC cell lines indicated that *VTCN1* positively correlated with a mucinous marker *MUC5B* while *PD-L1* negatively correlated with *HNF4A*, another mucinous marker (Fig 3D and E). A similar analysis using the RNA-seq dataset from the 230 TCGA LUAD cases indicated that *VTCN1* positively correlated with *HNF4A*, while *PD-L1* negatively

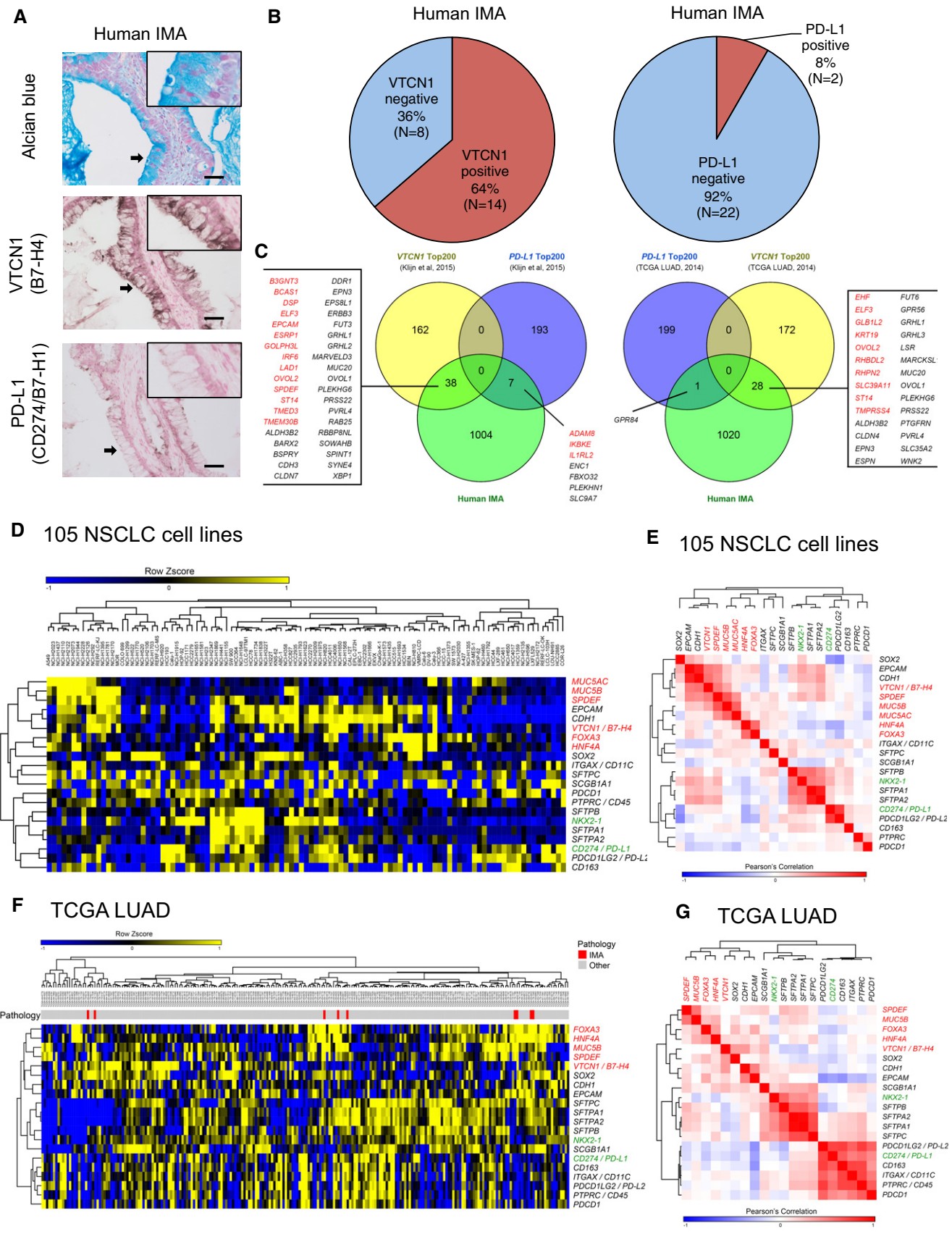

Figure 3.

◀

**Figure 3. VTCN1 but not PD-L1 is expressed in human IMA.**

A Shown is immunohistochemistry (IHC) detecting VTCN1 but not PD-L1 in human IMA. Alcian blue detects mucus. Scale bar: 50 μm. Insets show higher magnification of regions indicated by arrows.

B 64% of the human IMA expressed VTCN1 but most of the human IMA did not express PD-L1 as determined by IHC.

C Left panel, genes highly correlated with *VTCN1* or *PD-L1* in NSCLC cell lines (*n* = 105; Klijn *et al*, 2015) were assessed as to whether they are expressed in human IMA. 38 genes highly correlated with *VTCN1* were expressed in human IMA. However, only seven genes highly correlated with *PD-L1* were expressed in human IMA. Right panel, likewise, 28 genes highly correlated with *VTCN1* in the TCGA LUAD cases (*n* = 230; Cancer Genome Atlas Research Network, 2014) were expressed in human IMA but only 1 gene highly correlated with *PD-L1* was expressed in human IMA. Red indicates genes in the Mucinous Lung Tumor Signature (the 143 genes).

D Hierarchical clustering of the expression of IMA-related genes and cell type markers in the NSCLC cell lines (*n* = 105; Klijn *et al*, 2015). Genes in red color: pro-mucous genes. Genes in green color: anti-mucous genes.

E *VTCN1* but not *PD-L1* positively correlates with a mucous gene marker *MUC5B* in the NSCLC cell lines (*n* = 105; Klijn *et al*, 2015). Of note, the anti-mucous transcription factor *NKX2-1* was correlated with the pro-mucous transcription factor *SPDEF* in the 105 human NSCLC cell lines; however, *NKX2-1* was not correlated with *SPDEF* when 41 cell lines that lack the expression of both *NKX2-1* and *SPDEF* (RPKM ≤ 1) were excluded from the calculation, suggesting that the positive correlation was seen due to the large number of the cell lines that lack the expression of both *NKX2-1* and *SPDEF*. Genes in red color: pro-mucous genes. Genes in green color: anti-mucous genes.

F Hierarchical clustering of the expression of IMA-related genes and cell type markers in the TCGA LUAD cases (*n* = 230; Cancer Genome Atlas Research Network, 2014). Red indicates specimens with IMA pathology. Genes in red color: pro-mucous genes. Genes in green color: anti-mucous genes.

G *VTCN1* but not *PD-L1* positively correlates with a mucinous tumor marker *HNF4A* in the TCGA LUAD cases (*n* = 230; Cancer Genome Atlas Research Network, 2014). Overall, *VTCN1* but not *PD-L1* associates with mucous gene markers. Genes in red color: pro-mucous genes. Genes in green color: anti-mucous genes.

correlated with *FOXA3* and *SPDEF* (Fig 3F and G). These results suggest that VTCN1 is a better cancer immunotherapeutic target for human IMA than PD-L1.

### NKX2-1 induces PD-L1 in human mucinous lung cancer cells

In the analysis using the RNA-seq data from the 105 NSCLC cell lines, *PD-L1* positively correlated with *NKX2-1* (Fig 3E), a transcription factor absent in human IMA (Travis *et al*, 2011). Our ChIP-seq and cDNA microarray analyses indicated that ectopic NKX2-1 bound to the locus of *PD-L1/PD-L2* (Fig 4A and Appendix Fig S1) and induced the expression of *PD-L1* and *PD-L2* in mucus-producing A549 human lung carcinoma cells (Fig 4B; Maeda *et al*, 2012), which was also confirmed at the protein level in A549 cells and other mucus-producing lung cancer cell lines (Fig 4C and D). Next, we assessed whether PD-L1 is expressed in NKX2-1-expressing tumor cells in human NSCLC specimens (Dataset EV1). Among the NKX2-1-positive tumor cases, 29% (20 out of 68) expressed PD-L1 (Fig 4E and F). Among the NKX2-1-negative tumor cases, only 12% (nine out of 72) expressed PD-L1 (Fig 4E and F). These results indicate that NKX2-1 co-localizes with PD-L1 in a significant

portion of human NSCLC cells (two-tailed Fisher's exact test: *P*-value = 2.078E-02). Using immunohistochemistry, it has been shown that IMA lacks the expression of NKX2-1 (Travis *et al*, 2011). We found that 23% (27 out of 116) of human non-IMA cases expressed PD-L1 (Fig 4G and H), while only 8% (two out of 24) of human IMA cases expressed PD-L1 (Fig 4G and H, excluding non-mucinous tumor cells heterogeneously existing with mucinous tumor cells), suggesting that PD-L1 is rarely expressed in human IMA. In addition, *HNF4A* (a negative downstream gene of NKX2-1; Maeda *et al*, 2012) that is expressed in human IMA was negatively correlated with *NKX2-1* and *PD-L1* in the 105 NSCLC cell lines (Fig 3E) and the 230 TCGA LUAD cases (Fig 3G), further suggesting a negative association of PD-L1 with human IMA.

### FOXA3 or SPDEF along with KRAS[G12D] induces mucinous lung tumors *in vivo*

The Mucinous Lung Tumor Signature included the pro-mucous transcription factors *FOXA3* and *SPDEF* (Figs 1 and EV2, and Dataset EV3), which were among the top 50 genes highly induced in human IMA (Fig 5A and Dataset EV2). FOXA3 or SPDEF induces

**Figure 4. NKX2-1 induces PD-L1 in human mucinous lung cancer cell lines.** ▶

A A549 cells were infected with *Nkx2-1*-expressing lentivirus as previously reported (Maeda *et al*, 2012). ChIP-seq indicates that NKX2-1 bound to the locus of *PD-L1* and *PD-L2*.

B *PD-L1* and *PD-L2* mRNAs were significantly induced by NKX2-1. Results are expressed as mean ± SEM of biological triplicates for each group. *P* < 0.05 versus control was considered significant (Student's *t*-test). Gene expression was normalized by comparison with the constitutive expression of *GAPDH*. Control: control lentivirus; *Nkx2-1*: *Nkx2-1*-expressing lentivirus.

C A549 cells stably expressing *Nkx2-1* were developed as described in (A) and (B). Protein expression was confirmed by IB. PD-L1 was detected by antibodies from Cell Signaling (CST), Spring Bioscience (Spring) and Sino Biological (Sino) as described in Materials and Methods. ACTA1 was used as a loading control. Shown is a representative image from three independent experiments. NKX2-1 induced PD-L1 in A549 cells.

D H2122, Calu-3, and H292 mucus-producing lung cancer cells were infected with control lentivirus or *Nkx2-1*-expressing lentivirus. Protein expression was confirmed by IB. PD-L1 antibody from Cell Signaling (CST) was used to detect PD-L1 protein. ACTA1 was used as a loading control. Shown is a representative image from two independent experiments. NKX2-1 induced PD-L1 in the mucus-producing lung cancer cells.

E NKX2-1 (black) was expressed in the nucleus of non-mucinous lung tumor cells (non-IMA) but not in mucinous lung tumor cells (IMA) in human. Scale bar: 50 μm.

F PD-L1 was expressed in NKX2-1-positive lung tumor cells compared to NKX2-1-negative lung tumor cells in human (two-tailed Fisher's exact test: *P*-value = 2.078E-02). PD-L1-positive cases include > 5% of tumor cells expressing PD-L1 in cell membrane.

G PD-L1 (black) was expressed in cell membranes of non-mucinous lung tumor cells (non-IMA) but not in mucinous lung tumor cells (IMA) in human. Scale bar: 50 μm.

H PD-L1 was expressed in non-mucinous lung tumor cells (non-IMA) but not in mucinous lung tumor cells (IMA) in human. PD-L1-positive cases were determined as described in (F).

Source data are available online for this figure.

mucus-producing goblet cells in airway lung epithelial cells in transgenic mouse models (Park *et al*, 2007; Chen *et al*, 2014). FOXA3 was expressed in the nucleus of mucinous tumor cells in 86% (12 out of 14) of human IMA (Fig 5B and Dataset EV1). In conditional transgenic mouse models (Fig EV3A), induction of FOXA3 or SPDEF along with KRAS^G12D in lung epithelium resulted in the production

of mucinous lung tumors (Figs 5C and D, and EV3, and Dataset EV6). Although both transgenic mouse models developed mucinous lung tumors, induction of SPDEF along with KRAS^G12D produced malignant mucinous lung tumors (tubulopapillary-like carcinoma), while induction of FOXA3 along with KRAS^G12D produced benign mucinous lung tumors (e.g., papilloma). The induction of FOXA3 or

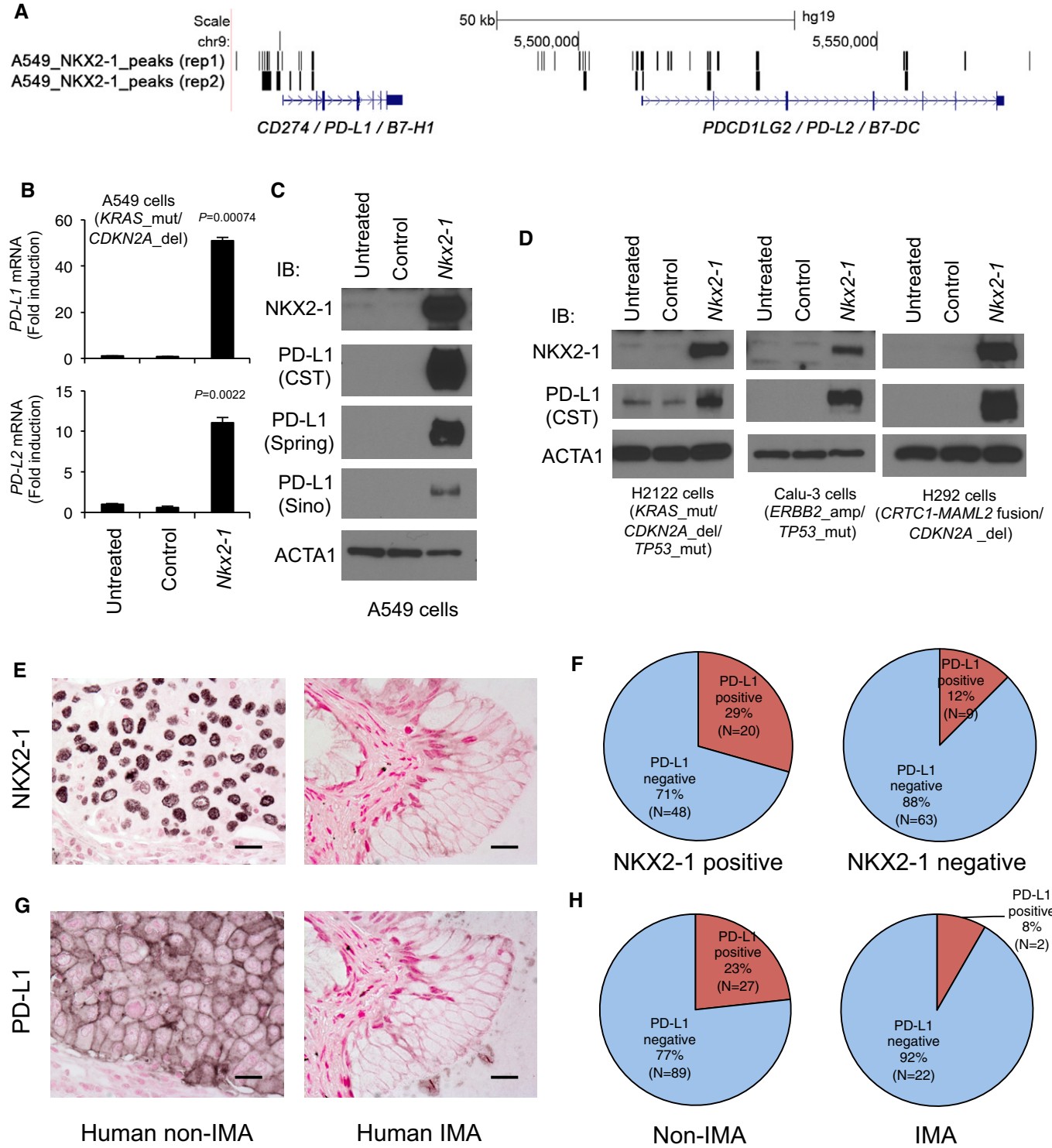

Figure 4.

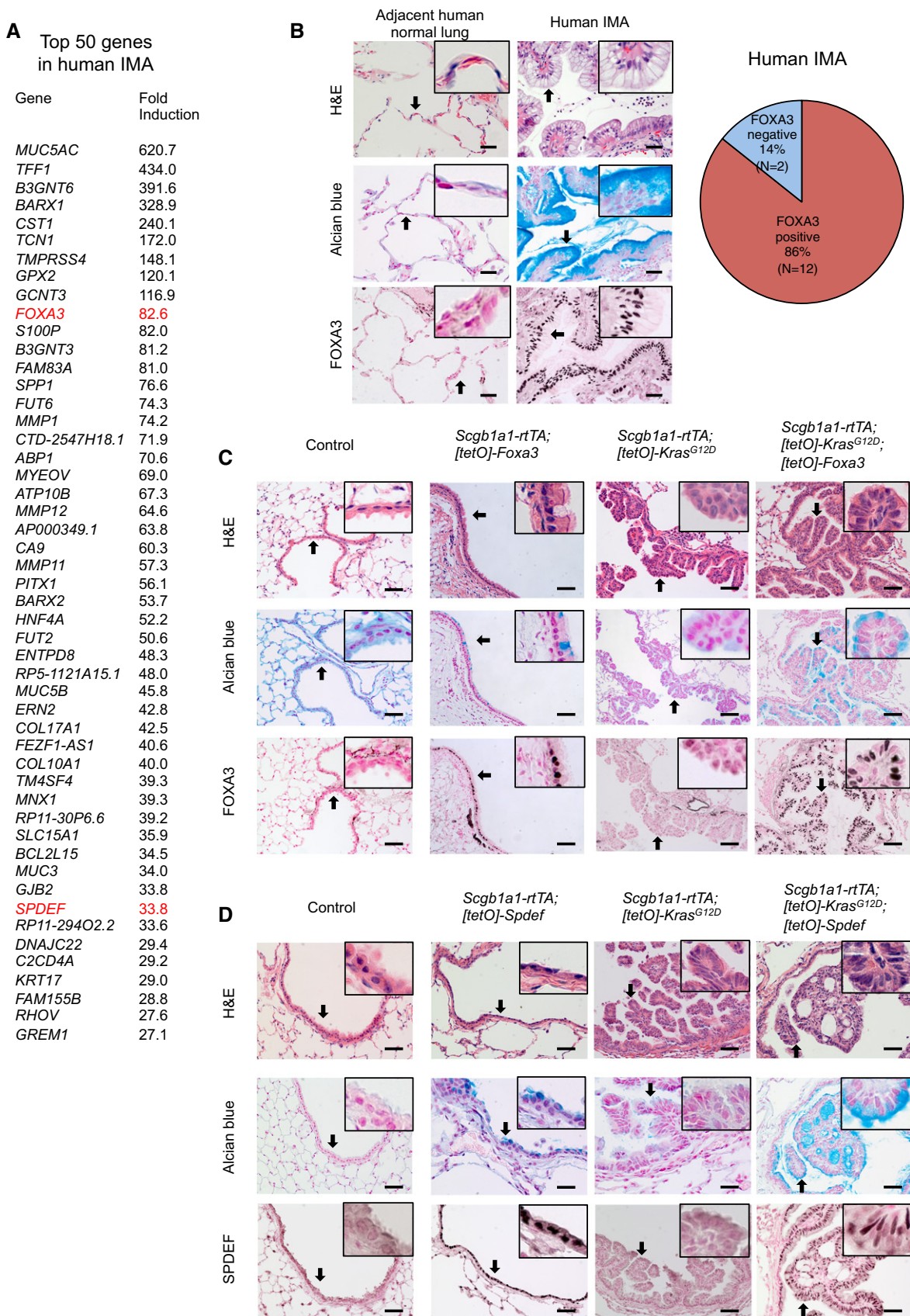

**Figure 5.**

◄

**Figure 5.   FOXA3 or SPDEF produces mucinous lung tumors in the presence of KRAS^G12D in transgenic mouse models.**

A   Shown are the top 50 genes highly expressed in human IMA. Transcription factors *FOXA3* and *SPDEF* (red) were highly expressed in human IMA.
B   Left panel, FOXA3 is expressed in the nucleus of human IMA but not in adjacent normal lung tissue. H&E: hematoxylin and eosin stain. Alcian blue detects mucus. Scale bar: 50 μm. Insets show higher magnification of regions indicated by arrows. Right panel, most of the human IMA express FOXA3.
C   KRAS^G12D and/or FOXA3 are induced in lung epithelium upon doxycycline administration in a tet-on transgenic mouse model (see Fig EV3 and Dataset EV6). Benign mucinous lung tumors (papilloma) were developed in the presence of both KRAS^G12D and FOXA3 around 6 months after doxycycline administration. H&E: hematoxylin and eosin stain. Alcian blue detects mucus. Scale bar: 50 μm. Insets show higher magnification of regions indicated by arrows.
D   KRAS^G12D and/or SPDEF are induced in lung epithelium upon doxycycline administration in a tet-on transgenic mouse model (see Fig EV3 and Dataset EV6). Malignant mucinous lung tumors (tubulopapillary-like carcinoma) were developed in the presence of both KRAS^G12D and SPDEF around 4 months after doxycycline administration. H&E: hematoxylin and eosin stain. Alcian blue detects mucus. Scale bar: 50 μm. Insets show higher magnification of regions indicated by arrows.

SPDEF without KRAS^G12D induced airway goblet cells but not mucinous lung tumors (Fig 5C and D and Dataset EV6), which is consistent with previous reports (Park *et al*, 2007; Chen *et al*, 2014). KRAS^G12D itself induced lung tumors but not mucinous lung tumors (Figs 5C and D, and EV3B, and Dataset EV6), which is also consistent with previous reports (Fisher *et al*, 2001; Maeda *et al*, 2012). These results suggest that FOXA3 or SPDEF along with mutant KRAS may be sufficient to induce mucinous lung tumors in human IMA.

**Anti-mucous NKX2-1 and pro-mucous FOXA3, SPDEF and HNF4A differentially regulate IMA-related genes in human lung cancer cells**

Next, we sought to determine whether FOXA3 or SPDEF recapitulates the *in vivo* function in human lung cancer cells that harbor a *KRAS* mutation. FOXA3 or SPDEF independently induced the mucin genes *MUC5AC* and *MUC5B*, biomarkers for IMA, in human A549 lung carcinoma cells that harbor a *KRAS* mutation (Fig 6A–C; Chen *et al*, 2014), suggesting that FOXA3 or SPDEF along with mutant KRAS intrinsically drives a mucinous phenotype in human IMA as

well. In addition to *MUC5AC* and *MUC5B*, differential gene expression analysis using RNA-seq identified 17 IMA-related genes induced by SPDEF and FOXA3 in common in A549 cells (Fig 6D, red and Dataset EV7; all of the genes shown in Fig 6D are genes induced in human IMA; see Dataset EV2).

HNF4A was previously identified as a marker for mouse and human IMA (Maeda *et al*, 2012; Snyder *et al*, 2013; Sugano *et al*, 2013); however, its regulation in human IMA is not well known. As we reported previously, HNF4A was repressed by NKX2-1 at the mRNA level (Maeda *et al*, 2012) and at the protein level as well (Fig 6A). Unexpectedly, HNF4A was repressed by SPDEF but induced by FOXA3 (Fig 6A). There is a group of such genes that are induced by FOXA3 but not by SPDEF (Fig 6D, green). Next, we sought to determine the function of HNF4A in A549 cells. HNF4A inhibited the expression of *MUC5AC* and *MUC5B* but induced *MUC3* (Fig 6E), which recapitulates a role of HNF4A in mouse intestine *in vivo* that was reported previously (Ahn *et al*, 2008). These results indicate that transcription factors expressed in human IMA differentially regulate the expression of IMA-related genes, including mucins, in part by recapitulating intestinal gene regulation (Figs 6E and EV4).

**Figure 6.   Anti-mucous (NKX2-1) and pro-mucous (FOXA3, SPDEF, and HNF4A) transcription factors differentially regulate mucin gene expression in human lung cancer cells.**

A   Lentivirus expressing *Nkx2-1* inhibited SPDEF, FOXA3, and HNF4A protein expression in A549 cells. SPDEF induced FOXA3 but inhibited HNF4A. FOXA3 induced SPDEF and HNF4A (three independent experiments). Constitutively expressed ACTA1 protein was used as a loading control. Endogenous SPDEF is marked by *.
B   *MUC5AC* and *MUC5B* mRNAs were inhibited by NKX2-1 and induced by SPDEF and FOXA3. Gene expression was normalized by comparison with the constitutive expression of *GAPDH*. Results are expressed as mean ± SEM of biological triplicates for each group. $P < 0.05$ versus control was considered significant (Student's *t*-test).
C   A549 cells were infected with control or *SPDEF*-expressing lentivirus. Control or SPDEF-expressing A549 cells were transfected with control or *FOXA3* siRNA. Three days after transfection, RNA was extracted from the A549 cells. The RNA was used for Taqman gene expression qPCR analysis as described in Materials and Methods. Gene expression was normalized by comparison with the constitutive expression of *GAPDH*. Results are expressed as mean ± SEM of experimental triplicates for each group. SPDEF-expressing lentivirus significantly induced mRNA expression of *SPDEF, FOXA3, MUC5AC*, and *MUC5B* in the presence of control siRNA ($P < 0.05$ was considered significant on *SPDEF* virus versus control virus; Student's *t*-test). *FOXA3* siRNA significantly reduced the endogenous expression of *FOXA3* mRNA (#$P < 0.05$ compared to control siRNA is considered significant). SPDEF significantly induced the expression of *MUC5AC* and *MUC5B* in the presence of *FOXA3* siRNA ($P < 0.05$ was considered significant on *SPDEF* virus versus control virus; Student's *t*-test). Two independent experiments were performed.
D   A549 cells expressing NKX2-1 (Maeda *et al*, 2012) were infected with control lentivirus or lentivirus expressing *SPDEF* or *Foxa3*. Differentially expressed genes from biological triplicates were analyzed by RNA-seq as described in Materials and Methods. Among the genes induced by SPDEF or FOXA3 in A549 cells, genes also expressed in human IMA are shown. Among the genes reduced by NKX2-1 in A549 cells (Maeda *et al*, 2012), genes also expressed in human IMA are also shown. Genes in red are induced by both SPDEF and FOXA3. Among the genes in red, *AGR2, BACE2, HKDC1, MUC5AC, MUC5B*, and *SPDEF* were also reduced by NKX2-1. Genes in green were induced by FOXA3 and reduced by NKX2-1.
E   Left panel, lentivirus expressing *Hnf4a* inhibited SPDEF and FOXA3 protein expression in A549 cells (two independent experiments). ACTA1 was used as a loading control. Endogenous SPDEF is marked by *. Right panel, *MUC5AC* and *MUC5B* mRNAs were inhibited by HNF4A. Gene expression was normalized by comparison with the constitutive expression of *GAPDH*. Results are expressed as mean ± SEM of biological triplicates for each group. *P*–value $< 0.05$ versus control was considered significant (Student's *t*-test). *MUC3* mRNA was induced by HNF4A (Taqman products were run on a gel due to no expression in control). The expression of *GAPDH* mRNA is shown as control.
F   Summary of mucin gene regulation by mucus-related transcription factors (anti-mucous transcription factor NKX2-1; pro-mucous transcription factors FOXA3, SPDEF and HNF4A) in non-IMA or IMA.

Source data are available online for this figure.

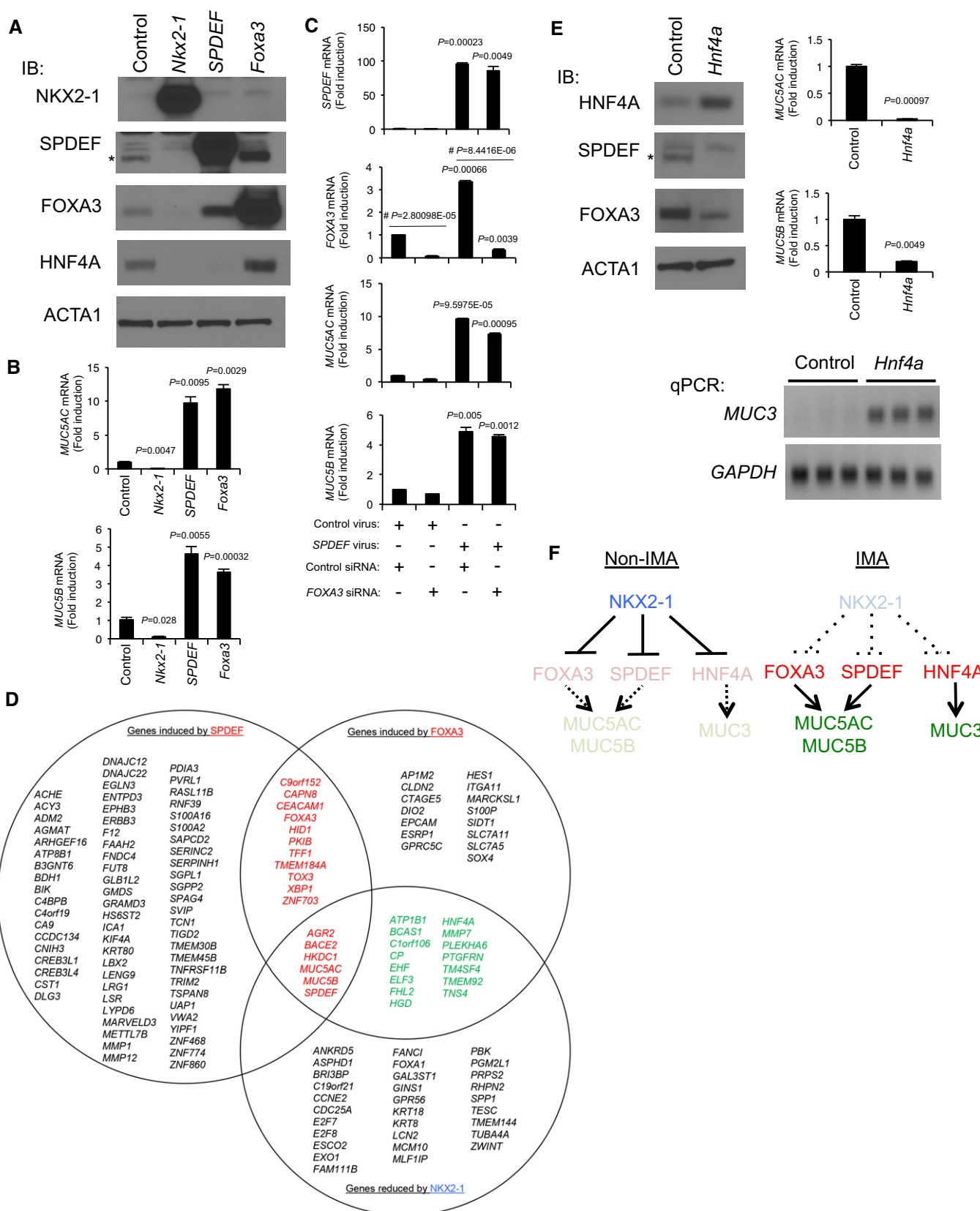

**Figure 6.**

**SPDEF binds to the functional enhancer regions in non-coding loci of *MUC5AC* and *MUC5B***

Previously, our ChIP-seq analyses determined that NKX2-1 and FOXA3 bound to the locus of *MUC5AC* in A549 lung carcinoma cells and BEAS-2B transformed bronchial epithelial cells (Maeda *et al*, 2012; Chen *et al*, 2014), suggesting that the expression of *MUC5AC* is directly regulated by NKX2-1 and FOXA3. However, it is unknown whether SPDEF directly binds to the loci of *MUC5AC* and/or *MUC5B* in lung carcinoma cells. Thus, we performed ChIP-seq to identify SPDEF binding sites in A549 cells (Figs 7, 8, and EV5). SPDEF bound to the non-coding upstream regions of *MUC5AC* and *MUC5B* (Figs 7A, 8A, and EV5), which is in part consistent with SPDEF binding sites identified by ChIP-seq using a MCF7 breast adenocarcinoma cell line (Figs 7A, 8A, and EV5; Fletcher *et al*, 2013). SPDEF binding regions on the *MUC5AC* and *MUC5B* loci were ~7 kb and ~3 kb away, respectively, from transcription start sites and distant from proximal promoter regions, including CRE and Sp1 binding sites (Figs 7B, 8B, and EV5; Choi *et al*, 2011; Kreda *et al*, 2012). Interestingly, the SPDEF binding site located in the ~3 kb upstream region of *MUC5B* contained rs35705950, a SNP that was previously reported as associated with idiopathic pulmonary fibrosis, in which the lung hypersecretes MUC5B (Fig 8B; Seibold *et al*, 2011). In order to determine whether the upstream regions bound by SPDEF are required for *MUC5AC* and *MUC5B* gene expression, we deleted these regions in A549 cells using CRISPR/Cas9 with two independent sgRNAs for each region (Figs 7A–C and 8A–C). The expression of *MUC5AC* in A549 cells that lack the upstream region (535 bp) of *MUC5AC* (Fig 7D, Deleted) was significantly reduced compared to that in control A549 cells (Fig 7D, Control; 94% reduction). The expression of *AGR2, FOXA3, MUC5B,* and *SPDEF* in the A549 (Fig 7D, Deleted) cells was similar to that in the A549 (Fig 7D, Control) cells. Likewise, the expression of *MUC5B* in A549 cells that lack the upstream region (746 bp) of *MUC5B* (Fig 8D, Deleted) was significantly reduced compared to that in control A549 cells (Fig 8D, Control; 87% reduction). The expression of *AGR2, FOXA3, MUC5AC,* and *SPDEF* in the A549 (Fig 8D, Deleted) cells was similar to that in the A549 (Fig 8D, Control) cells. These results indicate that the upstream regions bound by SPDEF contain functional enhancer elements required for the expression of *MUC5AC* and *MUC5B* in human IMA.

## Discussion

Using gene expression profiles from mouse models and human IMA specimens, we determined a Mucinous Lung Tumor Signature of genes for human IMA. Our analysis led to the identification of "druggable" molecular targets, including an immune checkpoint VTCN1, that are expressed in human IMA. We further determined that the transcription factor FOXA3 or SPDEF in the presence of mutant KRAS produced mucinous lung tumors *in vivo*. We investigated the molecular mechanisms by which mucin gene expression is regulated in human IMA and demonstrated that the upstream regions of *MUC5AC* and *MUC5B,* which are bound by SPDEF, were required for the expression of *MUC5AC* and *MUC5B* in A549 cells that harbor a *KRAS* mutation. The Mucinous Lung Tumor Signature and mouse models that we developed will be useful to identify

therapeutic targets and for preclinical testing of novel strategies to treat human IMA.

Multiple reports suggest that treatment success using therapeutic antibodies targeting an immune checkpoint PD-L1 or PD-1 depends on the expression of PD-L1 in lung tumors, including tumor cells and/or tumor-associated cells (Herbst *et al*, 2014; Garon *et al*, 2015; Hirsch *et al*, 2016). In our analysis using RNA-seq data from six human IMA specimens and immunohistochemistry from 24 human IMA specimens, PD-L1 was rarely expressed in human IMA (Figs 1, 3, and 4). We further investigated the expression of *PD-L1* at the mRNA level using RNA-seq data from the 105 human NSCLC cell lines (Klijn *et al*, 2015) and the 230 TCGA LUAD cases (Cancer Genome Atlas Research Network, 2014). We used these two datasets in order to compensate for the limitations of the individual datasets. The datasets from the 105 human NSCLC cell lines represent gene expression profiles of isolated lung tumor cells; however, these might be different from lung tumor cells in an intact tumor microenvironment since these cell lines were established after multiple passages on plastic dishes. The datasets from the 230 TCGA LUAD cases represent gene expression profiles of intact lung tumors; hence, they are the combined gene expression profiles of not only lung tumor cells but also tumor-associated fibroblasts, endothelial, and immune cells. This mixture of expression profiles from tumor and tumor-associated cells makes it difficult to assess whether the association of *PD-L1* with genes expressed in human IMA occurs in lung tumor cells and/or other tumor-associated cells. Thus, analysis using the two datasets provides a more comprehensive assessment of gene-to-gene correlation than that using only one dataset. In our analysis, *PD-L1* was not correlated with genes expressed in human IMA in either dataset compared to another immune checkpoint *VTCN1* (Fig 3). These results suggest that PD-L1 is rarely expressed in human IMA. Transcription factors such as STAT1, NF-κB, IRF1, STAT3, AP-1, and MYC have been shown to activate *PD-L1* (Liang *et al*, 2003; Loke & Allison, 2003; Lee *et al*, 2006; Marzec *et al*, 2008; Green *et al*, 2012; Casey *et al*, 2016); however, such transcription factors are ubiquitously expressed, which does not explain the mechanism by which PD-L1 is rarely expressed in human IMA. Our present data show that NKX2-1, which is absent in IMA (Travis *et al*, 2011), induced PD-L1 in mucinous lung cancer cells *in vitro* (Fig 4), suggesting that the lack of PD-L1 in IMA may be due to the absence of NKX2-1.

Contrary to the lack of expression of PD-L1 (B7-H1) in IMA, another immune checkpoint VTCN1 (B7-H4) was highly expressed in a portion of IMA at the mRNA and protein levels (Figs 1, 3, EV1, and EV2), which suggests that VTCN1 but not PD-L1 may be a better therapeutic immune checkpoint to target IMA. Though *VTCN1* was more correlated with mucous gene markers than *PD-L1* (Fig 3C), *VTCN1* was not regulated by the transcription factor SPDEF, FOXA3, or NKX2-1 in A549 lung carcinoma cells (Dataset EV7; Maeda *et al*, 2012; of note, A549 cells do not express endogenous VTCN1). SPDEF did not bind to the locus of *VTCN1* in A549 cells (Appendix Fig S2A), further suggesting that SPDEF does not regulate the expression of *VTCN1* in A549 cells. In addition, regardless of the expression of FOXA3, VTCN1 was expressed in human IMA (Appendix Fig S2B), also suggesting that FOXA3 may not regulate the expression of *VTCN1*. The transcriptional regulation of *VTCN1* is not well understood compared to that of *PD-L1*. Recently, HIF1A has been shown to induce *VTCN1* in multiple myeloma,

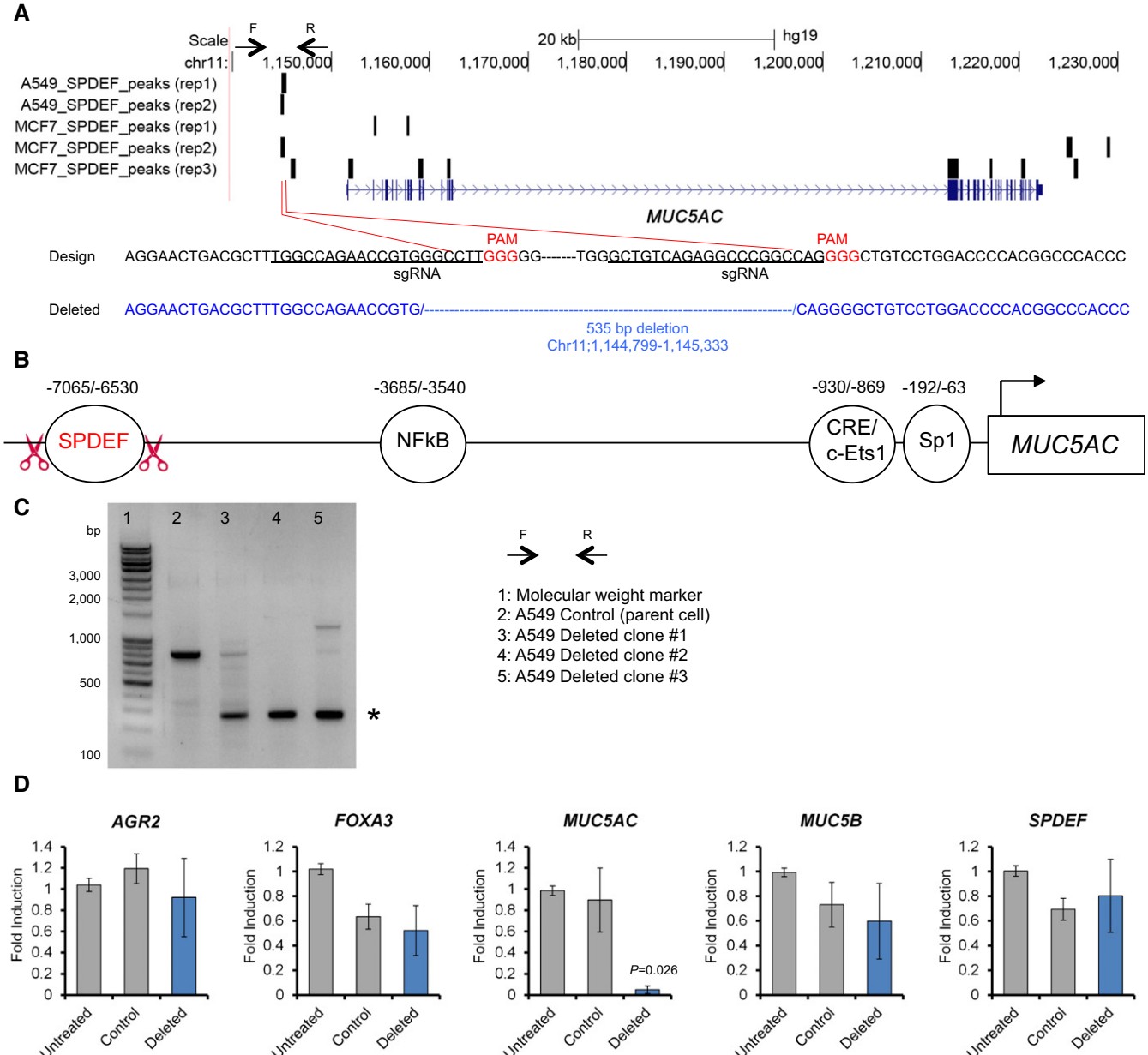

**Figure 7.  The upstream region of *MUC5AC* bound by SPDEF is required for the expression of *MUC5AC* in human lung cancer cells.**

A   Top panel, ChIP-seq was performed twice using SPDEF antibody and chromatin from A549 cells as described in Materials and Methods. The SPDEF binding site was identified in the upstream region of the *MUC5AC* gene in A549 lung carcinoma cells (duplicate) and MCF7 breast adenocarcinoma cells (triplicate). Chr, chromosome. F (forward) and R (reverse) arrows indicate the primer locations to amplify the genome region that is bound by SPDEF. Bottom panel, two sgRNAs were designed to delete the SPDEF binding region.

B   The non-coding upstream region of *MUC5AC* that harbors different transcription factor-binding sites. Red scissors indicate the locations where CRISPR/Cas9 made DNA double-strand breaks.

C   Three independent clones where precise deletion of the region (*) occurred. F (forward) and R (reverse) primers based on (A) were used to amplify the region that was bound by SPDEF.

D   Only *MUC5AC* mRNA, but not other mucus-related gene mRNAs, was significantly reduced in the three clones compared to the control (vector only) A549 cells. Gene expression was normalized by comparison with the constitutive expression of *GAPDH*. Results are expressed as mean ± SEM of biological replicates for each group. *P* < 0.05 versus control was considered significant (Student's *t*-test). Untreated, no transfection (7 clones); control, vector only (eight clones); deleted, vector with sgRNAs (three clones).

cervical, and breast cancer cell lines (Jeon *et al*, 2015), which may be a part of the mechanism by which *VTCN1* is regulated in IMA. As shown in Fig 3C, a transcription factor *ELF3*, which is expressed in

human IMA, was highly correlated with *VTCN1* in both the 105 human NSCLC cell lines (Klijn *et al*, 2015) and the 230 TCGA LUAD cases (Cancer Genome Atlas Research Network, 2014), suggesting

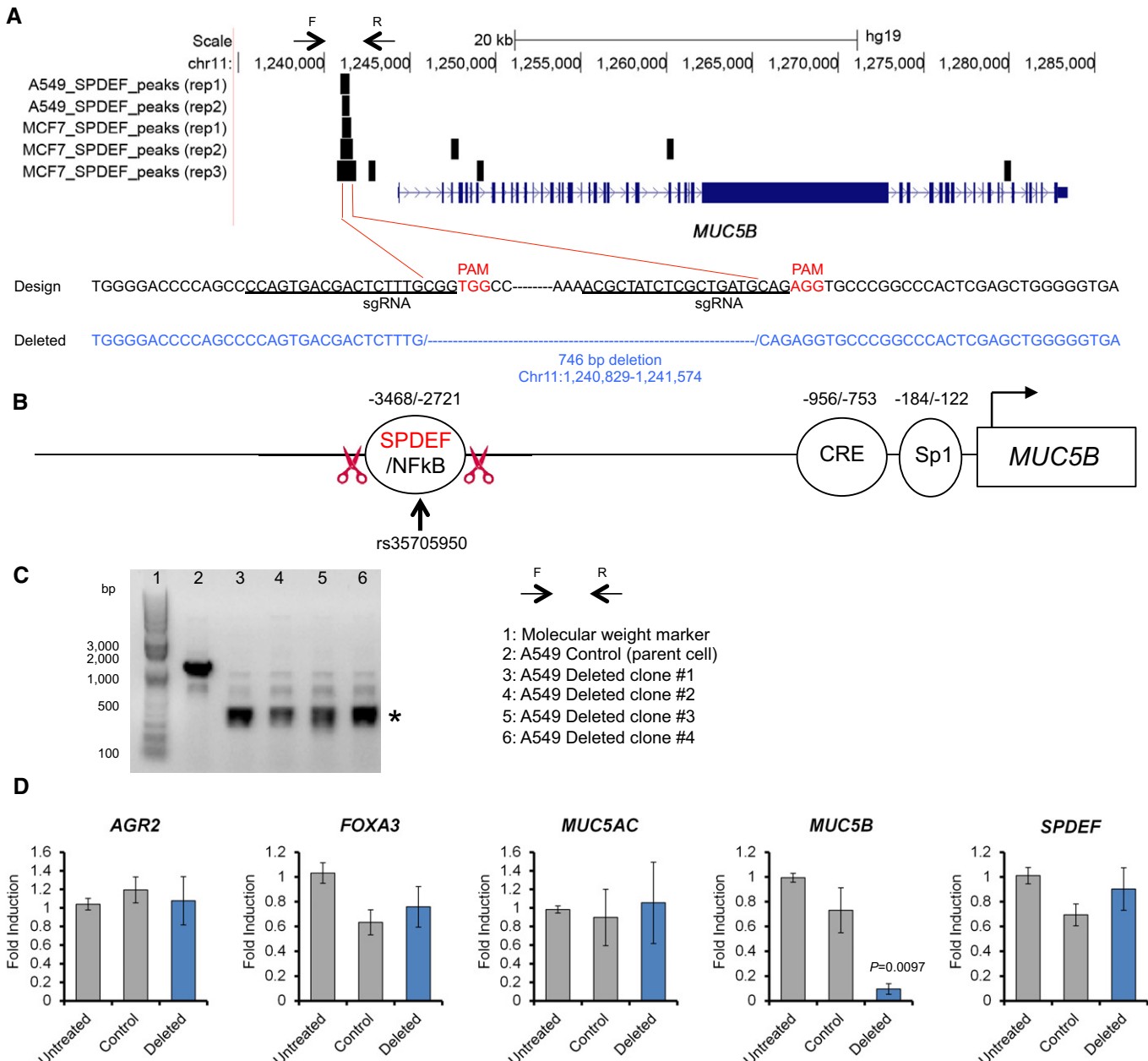

**Figure 8. The upstream region of *MUC5B* bound by SPDEF is required for the expression of *MUC5B* in human lung cancer cells.**

A  Top panel, ChIP-seq was performed twice using SPDEF antibody and chromatin from A549 cells as described in Materials and Methods. The SPDEF binding site was identified in the upstream region of the *MUC5B* gene in A549 lung carcinoma cells (duplicate) and MCF7 breast adenocarcinoma cells (triplicate). Chr, chromosome. F (forward) and R (reverse) arrows indicate the primer locations to amplify the genome region that is bound by SPDEF. Bottom panel, two sgRNAs were designed to delete the SPDEF binding region.

B  The non-coding upstream region of *MUC5B* that harbors different transcription factor-binding sites. Red scissors indicate the locations where CRISPR/Cas9 made DNA double-strand breaks. The SNP rs35705950 is seen in idiopathic pulmonary fibrosis patients who hypersecrete MUC5B (Seibold *et al*, 2011).

C  Four independent clones that have precise deletion of the region (*) were obtained. F (forward) and R (reverse) primers based on (A) were used to amplify the region that was bound by SPDEF.

D  Only *MUC5B* mRNA, but not other mucus-related gene mRNAs, was significantly reduced in the four replicate clones compared to the control (vector only) A549 cells. Gene expression was normalized by comparison with the constitutive expression of *GAPDH*. Results are expressed as mean ± SEM of biological replicates for each group. *P* < 0.05 versus control was considered significant (Student's *t*-test). Untreated, no transfection (seven clones); control, vector only (eight clones); deleted, vector with sgRNAs (four clones).

that ELF3 may be a regulator of *VTCN1*. Further studies using additional IMA cell lines other than A549 cells are required to elucidate the mechanism by which *VTCN1* is regulated in IMA.

In the present study, we determined that SPDEF or FOXA3 along with mutant KRAS induces autochthonous mucinous lung tumors in transgenic mice. SPDEF has been reported to be overexpressed and

to promote tumor growth in breast, ovarian, and prostate cancers (Ghadersohi & Sood, 2001; Gunawardane *et al*, 2005; Rodabaugh *et al*, 2007; Sood *et al*, 2007; Buchwalter *et al*, 2013), while para-doxically SPDEF has also been reported to function as a tumor suppressor in breast, ovarian, prostate, and colon cancers (Feldman *et al*, 2003; Gu *et al*, 2007; Ghadersohi *et al*, 2008; Turner *et al*, 2008; Noah *et al*, 2013; Cheng *et al*, 2014). The inconsistent results in breast, ovarian, and prostate cancers might be the result of using various models with different tumor cell types harboring distinct tumorigenic pathways in those cancer groups. In our study, *SPDEF* was highly expressed in human IMA (Figs 1 and EV1). Transgenic mice that conditionally express SPDEF along with mutant KRAS (KRAS$^{G12D}$) in lung epithelium had a shorter survival time than the mice that express only KRAS$^{G12D}$ (Fig EV3C, right panel). SPDEF along with KRAS$^{G12D}$ induced malignant mucinous lung tumors (tubulopapillary-like carcinoma; Fig EV3B), while KRAS$^{G12D}$ itself induced benign non-mucinous lung tumors (Fig EV3B). These results indicate that SPDEF promotes mutant KRAS lung tumorigenesis. On the contrary, FOXA3 was reported to inhibit the growth of hepatocellular carcinoma cells *in vitro* (Ding *et al*, 2014). In our study, FOXA3 was highly expressed in human IMA (Figs 1, 5, and EV1); however, transgenic mice that conditionally express FOXA3 along with KRAS$^{G12D}$ in lung epithelium survived longer than the mice that express only KRAS$^{G12D}$ (Fig EV3C). FOXA3 along with KRAS$^{G12D}$ induced benign mucinous lung tumors (Fig EV3B). These results suggest that FOXA3 itself may not promote the progression of mutant KRAS lung tumors. Although either SPDEF or FOXA3 along with mutant KRAS induces mucinous lung tumors *in vivo*, their roles in mutant KRAS-driven lung tumorigenesis (e.g., inva-siveness) might be different. Further studies including development of transgenic mice expressing both SPDEF and FOXA3 simultane-ously along with mutant KRAS may be required to determine the *in vivo* function of SPDEF and FOXA3 in human IMA. In addition, our study shows the importance of assessing IMA in a transgenic mouse model that develops autochthonous mucinous lung tumors in an intact lung structure rather than in a xenograft model in which the consequences of tumor growth and invasion on lung function cannot be assessed.

To our knowledge, there are at least seven different types of mouse models that develop mucinous lung tumors: (i) *Kras$^{G12D}$/Cdkn2a$^{-/-}$* (Fisher *et al*, 2001); (ii) transgenic *Tle1* (Allen *et al*, 2006); (iii) *Kras$^{G12D}$/Nkx2-1$^{+/-}$* or *Kras$^{G12D}$/Nkx2-1$^{flox/flox}$* (Maeda *et al*, 2012; Snyder *et al*, 2013); (iv) *Kras$^{G12D}$/Runx3$^{flox/flox}$* (Lee *et al*, 2013); (v) *Kras$^{G12D}$/Tp53$^{flox/flox}$/Eed$^{flox/flox}$* (Serresi *et al*, 2016); (vi) *Kras$^{G12D}$/transgenic Foxa3* (present study); and (vii) *Kras$^{G12D}$/transgenic Spdef* (present study). In human IMA, the co-occurrence of *KRAS* mutation and inactivation of *CDKN2A* is frequently seen (Fig 1D and E; Skoulidis *et al*, 2015), which suggests that the *Kras$^{G12D}$/Cdkn2a$^{-/-}$* (Fisher *et al*, 2001) model indeed recapitulates human IMA pathogenesis; however, the mecha-nism by which this co-occurrence induces mucinous lung tumors is not known. The model with reduced expression of *Nkx2-1* along with a *Kras* mutation, which recapitulates the human IMA patho-genesis, was independently developed by ourselves and others (Maeda *et al*, 2012; Snyder *et al*, 2013). We induced mutant KRAS in lung epithelial cells of *Nkx2-1* heterozygous mice by a tet-on system using doxycycline (Maeda *et al*, 2012), which recapitulates a genetic alteration seen in some human IMA cases (*KRAS* mutation

and *NKX2-1* heterozygous loss but not *NKX2-1* amplification; Fig 1D and E), while others induced mutant KRAS in lung cells (probably not specific to lung epithelial cells due to the adenovirus-mediated induction) and reduced NKX2-1 (*Nkx2-1$^{flox/flox}$*) by a Cre-loxP system using intratracheal injection of adenovirus carrying Cre (Snyder *et al*, 2013). The difference in gene expression profiles between the two model systems (Figs 1B and EV2) may be derived from the different mouse model systems. Nevertheless, the gene expression profile of either mouse model did not fully overlap with that of human IMA (Figs 1B and EV2), indicating the limitation of these mouse models. However, in order to leverage both mouse models, we included genes induced in both mouse models (along with the genes induced in human IMA) as the signature genes. These results indicate that additional mouse models are required to model human IMA. Our present models expressing FOXA3 or SPDEF along with mutant KRAS also recapitulate human IMA pathogenesis (Figs 5 and 6), collectively suggesting that airway goblet cells that are proficient in SPDEF and FOXA3 and deficient in NKX2-1 (Chen *et al*, 2009) may acquire a *KRAS* mutation, which in turn may trans-form the goblet cells into mucinous lung tumor cells. Induction of *TLE1* or reduction of *RUNX3* or *EED* was not seen in our RNA-seq data using human IMA (Dataset EV2), suggesting other mouse models may not directly recapitulate human IMA pathogenesis; however, the expression level of TLE1, RUNX3, and/or EED at the protein level may be critical in human IMA.

ChIP-seq has been used to identify transcription factor-binding sites and potential enhancer/silencer regions; however, it does not always indicate the functional significance of the sites/elements unless the site/elements are genetically modified or deleted at a genomic level. Such modification/deletions to determine their signif-icance in human cancer cells are now technically feasible using CRISPR/Cas9. For example, CRISPR/Cas9-mediated deletion (191 bp) of −7.5 kb upstream regions of *TAL1*, which is bound by the transcription factor MYB, reduced the expression of *TAL1* by ~85% in human Jurkat leukemia cells (Mansour *et al*, 2014). CRISPR/Cas9-mediated deletion (1,555–1,727 bp) of a part of the super-enhancer region of *MYC*, which is bound by the transcrip-tional co-activator EP300, reduced the expression of *MYC* by ~30% in human H2009 lung adenocarcinoma cells (Zhang *et al*, 2016). CRISPR/Cas9-mediated deletion (4–13 bp) of the insulator CTCF binding site in the *PDGFRA* locus induced the expression of *PDGFRA* by 1.6-fold in human glioblastoma cells (Flavahan *et al*, 2016). These studies indicate that the degree of significance of the enhancer/insulator regions influencing the expression of associated genes is varied. In our study, CRISPR/Cas9-mediated deletions (535 or 746 bp) of −7 kb or −3.5 kb upstream regions of *MUC5AC* or *MUC5B*, which was bound by the transcription factor SPDEF, reduced the expression of *MUC5AC* or *MUC5B* by ~90% in human A549 lung carcinoma cells (Figs 7 and 8), which indicates that the upstream enhancer regions are indispensable for the expression of *MUC5AC* or *MUC5B* in human IMA. SPDEF binding to the upstream regions of *MUC5AC* and *MUC5B* was also shown by ChIP-seq in a MCF7 breast adenocarcinoma cell line (Figs 7A, 8A, and EV5C and D; Fletcher *et al*, 2013). MUC5AC and/or MUC5B are highly induced in breast and pancreatic cancers (Sónora *et al*, 2006; Kaur *et al*, 2013), and MUC5B was shown to promote tumorigenesis of breast cancer (Valque *et al*, 2012). Genetic and/or epigenetic analysis of the upstream enhancer regions in breast and pancreatic cancers,

which were functionally validated in this study, may lead to the understanding of the mechanism by which MUC5AC and/or MUC5B is induced in those cancers.

In summary, we identified a novel gene signature for human IMA, which includes therapeutically targetable genes. Our two novel mouse models, in addition to previous models of IMA, will be useful to test potential therapeutics *in vivo*. We identified critical enhancers that are required for the gene expression of two major mucins in human IMA, which may also provide the mechanisms of mucin hypersecretion in other cancers such as gastrointestinal, pancreatic, and breast cancers and chronic lung diseases, including idiopathic pulmonary fibrosis (IPF), cystic fibrosis (CF), chronic obstructive pulmonary disease (COPD), and asthma.

# Materials and Methods

## Human specimens

RNA samples for invasive mucinous adenocarcinoma of the lung (IMA) and adjacent normal lung tissues from six patients who were enrolled in this study were obtained in accordance with institutional guidelines for use of human tissue for research purposes from the Lungen-Biobank Heidelberg, Germany, which is a member of the Biomaterial Bank Heidelberg (BMBH) and the biobank platform of the German Centre for Lung Research (DZL; approval # S 270/2001 V2.0). Paraffin sections for invasive mucinous (IMA; $n = 24$) and non-mucinous (non-IMA; $n = 116$) adenocarcinomas of the lung were obtained in accordance with institutional guidelines for use of human tissue for research purposes from the University of Cincinnati Cancer Institute Tumor Bank, Cincinnati, Ohio (approval # TB0049), Japan; Kawasaki Medical School, Okayama, Japan (approval # 1310); Nagasaki University Graduate School of Biomedical Sciences, Nagasaki, Japan (approval # 05062433-2); and US Biomax (Rockville, MD). Written informed consent was obtained from all participants. Patients' information is summarized in Dataset EV1.

## RNA-seq analysis of human specimens

RNA-seq was performed at the CCHMC DNA Sequencing and Genotyping Core. RNA-sequencing libraries were produced using the Illumina TruSeq preparation kit and subsequently sequenced on the Illumina HiSeq 2500, using paired-end, 100-bp reads. Data analysis was performed in GeneSpring NGS. First, sequenced reads were aligned to the human genome build hg19. Only high-quality reads were included in the analysis, following removal of multiply-mapped reads, reads with base quality ≤ 30, and zero "N's" allowed per read. Aligned reads were quantified in order to produce FPKM (fragments per kilobase of transcript per million mapped reads) for each transcript in each sample. Further, FPKM values were normalized using the quantile normalization procedure and baselined to the median of all samples. A final filtration was applied, requiring ≥ 10 reads in 100% of samples in at least one of the experimental groups. Expression values of each transcript were compared between IMA and adjacent normal lung tissues using a paired *t*-test in order to remove maximum inter-individual variations. Differentially expressed genes ($n = 2,621$) were identified using the

following criteria: Benjamini–Hochberg-adjusted $P < 0.05$ and fold change requirement of 2.0. Clustering analysis of differentially expressed genes was performed in GENE-E (http://www.broadinstitute.org/cancer/software/GENE-E/) using hierarchical clustering algorithm with Pearson's correlation-based distance and complete linkage.

## Gene Set Enrichment Analysis

In the GSEA analyses (Figs 1 and 2), the identified Mucinous Lung Tumor Signature was used as a gene set. Genes with normalized expression greater than 1 in at least 1 sample in the datasets were selected as the background. Permutation type was "phenotype". The number of permutations was set to 1,000. "Signal2Noise" was used for ranking genes, and the enrichment statistic was "weighted". In the GSEA analysis of the signature using The Cancer Genome Atlas (TCGA) human lung adenocarcinoma (LUAD) data (Fig 1C), nine human IMA cases (Dataset EV4) were identified based on the "Pathology" annotation in the supplemental table of the original paper (Cancer Genome Atlas Research Network, 2014). For comparison, we randomly selected nine normal lung cases (Dataset EV4) from the available TCGA LUAD normal cases since the RNA-seq data of the adjacent normal lung tissue with eight out of nine human IMA cases were not available. The RNA-seq data of the human IMA and normal lung cases were downloaded through the Genomic Data Commons (GDC) Legacy Archive (https://gdc-portal.nci.nih.gov/legacy-archive) using the following filter: Center Name is "University of North Carolina", Data Category is "Gene Expression", and Data Type is "Gene Expression Quantification". The expression was quantile-normalized and log2-transformed. The minimum of the log2-transformed expression was set to 0. In the GSEA analysis of the signature using the human cancer cell line data (Fig 2B), 598 cell lines of 20 tissue groups and diseases (Dataset EV4) were selected based on the "Characteristics[tissue supergroup]" and "Characteristics[disease]" annotations in the supplemental table of the original paper (Klijn *et al*, 2015). The RNA-seq data of selected cell lines were downloaded from ArrayExpress using accession code E-MTAB-2706 (https://www.ebi.ac.uk/arrayexpress/experiments/E-MTAB-2706). The expression values were quantile-normalized using the "normalize.quantiles" function in the Bioconductor package preprocessCore (http://bioconductor.org/packages/release/bioc/html/preprocessCore.html) and log2-transformed. The minimum of the log2-transformed expression was set to 0.

## Analysis of TCGA-human lung adenocarcinoma (LUAD) samples ($n = 230$)

The RNA-seq data of human LUAD samples ($n = 230$) were downloaded from the TCGA data portal (https://tcga-data.nci.nih.gov/docs/publications/luad_2014/tcga.luad.rnaseq.20121025.csv.zip). Gene expression was quantified using RSEM (Li & Dewey, 2011) and normalized within-sample to a fixed upper quartile. Expression values of zero were set to the overall minimum value, and all data were log2-transformed. For details on the original processing of the data, refer to the supplemental information in the original paper (Cancer Genome Atlas Research Network, 2014). The correlations of genes with *VTCN1* (*B7-H4*) and *CD274* (*PD-L1/B7-H1*) were measured using Pearson's correlation. Genes expressed

(log2-transformed expression > 0) in at least 10% of LUAD samples were included in the correlation analysis. The clustering analyses of the expression and correlation of the selected cell type markers in LUAD samples (Fig 3F and G) were performed in GENE-E (http://www.broadinstitute.org/cancer/software/GENE-E/) using hierarchical clustering algorithm with Pearson's correlation-based distance and complete linkage. The clustering analyses of the expression of the Mucinous Lung Tumor Signature genes ($n$ = 141; *MUC5AC* and *MUC3A/B* were not detected in the 230 TCGA LUAD dataset) in all the 230 TCGA LUAD samples (Fig 1D) and in the nine IMA samples (Fig 1E) were performed in GENE-E (http://www.broadinstitute.org/cancer/software/GENE-E/) using hierarchical clustering algorithm with Pearson's correlation-based distance and complete linkage.

### Analysis of human non-small cell lung cancer (NSCLC) cell lines ($n$ = 105)

The RNA-seq data of human cancer cell lines ($n$ = 675) were downloaded from ArrayExpress using accession code E-MTAB-2706. 105 out of 675 samples were selected as non-small cell lung cancer (NSCLC) cell lines for analysis based on the following sample annotations; "OrganismPart" is lung and "Disease" is lung carcinoma, lung adenocarcinoma, lung anaplastic carcinoma, non-small cell lung carcinoma, squamous cell lung carcinoma, large cell lung carcinoma, lung mucoepidermoid carcinoma, lung papillary adenocarcinoma, lung adenosquamous carcinoma, bronchioloalveolar adenocarcinoma, or squamous cell carcinoma. Correlation and clustering analyses in Fig 3 were performed using variance-stabilized gene expression data downloaded from ArrayExpress using accession code E-MTAB-2706. For details on the original processing of the data, refer to the original paper (Klijn *et al*, 2015). The correlations of genes with *VTCN1* (*B7-H4*) and *CD274* (*PD-L1/B7-H1*) were measured using Pearson's correlation. The clustering analyses of the expression and correlation of the selected cell type markers in NSCLC cell lines (Fig 3D and E) were performed in GENE-E (http://www.broadinstitute.org/cancer/software/GENE-E/) using the hierarchical clustering algorithm with Pearson's correlation-based distance and complete linkage.

### Analysis of the expression of the Mucinous Lung Tumor Signature genes in 20 groups of human cancer cell lines

In the analysis of the expression of the Mucinous Lung Tumor Signature genes in different cancer types (Fig 2A), 598 cell lines of 20 tissue groups and diseases (Dataset EV4) were selected based on the "Characteristics[tissue supergroup]" and "Characteristics[disease]" annotations in the supplemental table of the original paper (Klijn *et al*, 2015). The RNA-seq data of selected cell lines were downloaded from ArrayExpress using accession code E-MTAB-2706 (https://www.ebi.ac.uk/arrayexpress/experiments/E-MTAB-2706). Expression values were measured in RPKM. For details on the original processing of the data, refer to the original paper (Klijn *et al*, 2015). The expression values were quantile-normalized using the "normalize.quantiles" function in the Bioconductor package preprocessCore (http://bioconductor.org/packages/release/bioc/html/preprocessCore.html) and log2-transformed. The minimum of the log2-transformed expression was set to 0. The expression of the signature

genes in a group of the cell lines was measured by the mean normalized expression of the gene in all the cell lines in the group. The clustering analysis of the expression of the signature genes ($n$ = 143) in 20 different cell line groups (Fig 2A) was performed in GENE-E (http://www.broadinstitute.org/cancer/software/GENE-E/) using hierarchical clustering algorithm with Pearson's correlation-based distance and average linkage.

### Histology and immunohistochemistry

Staining (H&E, Alcian blue and immunohistochemistry) was performed using 5-μm paraffin-embedded lung sections as previously described (Chen *et al*, 2009) except EDTA antigen retrieval was performed for PD-L1 staining. Antibodies used were rabbit anti-VTCN1 (1:100; clone D1M8I; cat# 14572, Cell Signaling Technology, Danvers, MA; validated by immunoblotting; see Appendix Fig S3), rabbit anti-PD-L1 (1:100; clone E1L3N; cat# 13684, Cell Signaling Technology), goat anti-FOXA3 (1:100; cat# sc-5361, Santa Cruz Biotechnology), mouse anti-NKX2-1/TTF-1 (1:500; cat# WRAB-1231; Seven Hills Bioreagents, Cincinnati, OH), and guinea pig anti-SPDEF (1:500; a gift from J. Whitsett; Park *et al*, 2007). Blind assessment of staining was performed by three investigators (K.T., I.M.F-B., and Y.M.).

### Cell culture, Taqman gene expression analysis, lentivirus infection, siRNA transfection, RNA-seq, and immunoblotting (IB)

A549 lung carcinoma cells (Lot# F-10600), H2122 lung adenocarcinoma cells (Lot# 59399669), Calu-3 lung adenocarcinoma cells (Lot# 57814093), and H292 lung mucoepidermoid carcinoma cells (Lot# 3895200) were obtained directly from ATCC (Manassas, VA). Mycoplasma contamination was not detected by the Universal Mycoplasma Detection Kit (cat # 30-1012, ATCC). A549 cells expressing NKX2-1, SPDEF, or FOXA3 were made using lentiviral vectors as previously described (Chen *et al*, 2009, 2014; Maeda *et al*, 2012). Transfection of siRNAs (cat# 4390843 for Negative Control #1, cat# 4390846 for Negative Control #2, and cat# 4392420 for *FOXA3* siRNA [s6695] and *VTCN1* siRNA [#1, s36082; #2, s36083; #3, s36084], Thermo Fisher Scientific, Waltham, MA) was performed using Lipofectamine® RNAiMAX Transfection Reagent (cat# 13778075, Thermo Fisher Scientific). RNA was extracted using Trizol Reagent (cat# 15596018, Thermo Fisher Scientific) and purified using RNeasy MinElute Cleanup Kit (cat# 74204, Qiagen, Valencia, CA). Taqman gene expression analysis with high-quality RNA was performed as previously described (Maeda *et al*, 2011) with Taqman probes (Hs00873651_mH for *MUC5AC*; Hs00861595_m1 for *MUC5B*; Hs03649367_mH for *MUC3A/B*; Hs00171942_m1 for *SPDEF*; Hs00270130_m1 for *FOXA3*; Hs00230853_m1 for *HNF4A*; Hs00968940_m1 for *NKX2-1*; Hs01552471_g1 for *VTCN1*; Hs01125301_m1 for *PD-L1* [*CD274*]; Hs00180702_m1 for *AGR2*; Hs99999903_m1 for *ACTB;* and Hs02758991_g1 for *GAPDH*). Number of replicates is described in legends. RNA-seq was performed at the CCHMC DNA Sequencing and Genotyping Core. RNA-sequencing libraries were produced using the Illumina TruSeq preparation kit and subsequently sequenced on the Illumina HiSeq 2500, using single-end, 75-bp reads. For each read, five left and five right end bases were trimmed. Reads were further trimmed if more than 10% of bases with a quality score < 20. The remaining reads

were aligned to the human genome build hg19. Differential expression analysis of RNA-seq was performed using DESeq2 (Love *et al*, 2014) with default parameterization. Differentially expressed genes were identified using the following criteria: Benjamini–Hochberg-adjusted $P < 0.05$ and fold change requirement of 2.0. A549 cells expressing HNF4A were made by infecting *Hnf4a*-expressing lentiviral vector made by inserting rat *Hnf4* cDNA (a gift from F. Sladek at the University of California, Riverside, CA) into the PGK-IRES-EGFP lentiviral vector as previously described (Maeda *et al*, 2011). Immunoblot assays were performed using rabbit anti-NKX2-1/TTF-1 (1:5,000, cat# WRAB-1231; Seven Hills Bioreagents, Cincinnati, OH), rabbit anti-SPDEF (1:5,000, cat# sc-67022X; Santa Cruz Biotechnology), goat anti-FOXA3 (1:5,000, cat# sc-5361X, Santa Cruz Biotechnology), goat anti-HNF4A (1:5,000, cat# sc-6556X, Santa Cruz Biotechnology), rabbit anti-PD-L1 antibody (1:5,000; clone E1L3N; cat# 13684, Cell Signaling Technology; cat# M4420, Spring Bioscience, Pleasanton, CA or cat# 10084-R015-50, Sino Biological, Beijing, China), rabbit anti-VTCN1 (1:5,000; clone D1M8I; cat# 14572, Cell Signaling Technology), and rabbit anti-ACTA1 (1:5,000; cat# A2066, Sigma-Aldrich) as described previously (Maeda *et al*, 2006). Statistical differences were determined using Mann–Whitney test or Student's *t*-test (two-tailed and unpaired). The difference between two groups was considered significant when the *P*-value was < 0.05.

## Mice

*[tetO]-Foxa3* and *[tetO]-Spdef* mice were provided by J. Whitsett at Cincinnati Children's Hospital Medical Center and the University of Cincinnati College of Medicine, Cincinnati, OH (Park *et al*, 2007; Chen *et al*, 2014) and crossed with *Scgb1a1-rtTA;[tetO]-Kras4b^{G12D}* (Maeda *et al*, 2012) to develop *Scgb1a1-rtTA;[tetO]-Kras4b^{G12D};[tetO]-Foxa3* and *Scgb1a1-rtTA;[tetO]-Kras4b^{G12D};[tetO]-Spdef* mice. All mice are FVB/N background. Transgenic mice were provided chow containing doxycycline (625 mg/kg chow) beginning at 4–5 weeks of age. Mouse maintenance and procedures were done in accordance with the institutional protocol guidelines of Cincinnati Children's Hospital Medical Center (CCHMC) Institutional Animal Care and Use Committee. Mice were housed in a pathogen-free barrier facility in humidity and temperature-controlled rooms on a 12:12-h light/dark cycle and were allowed food and water *ad libitum*. Since all transgenic mice were healthy before doxycycline administration, all of them were enrolled for the study. For the survival study, at least eight mice in each genotype were enrolled. For the histological study, at least five mice in each genotype that develop lung tumors were enrolled (at least three mice in each genotype that did not develop lung tumors were enrolled to minimize animal use). See Dataset EV6 for further mouse information.

## ChIP-seq and ChIP-qPCR

ChIP-seq was performed twice as previously described (Maeda *et al*, 2012) using A549 cells expressing NKX2-1 and/or SPDEF by lentivirus with rabbit anti-NKX2-1/TTF-1 (cat# WRAB-1231; Seven Hills Bioreagents) or rabbit anti-SPDEF (cat# sc-67022X; Santa Cruz Biotechnology; previously used for ChIP-seq by Fletcher *et al*, 2013). Illumina-sequencing libraries were prepared as previously described (Maeda *et al*, 2012). DNA sequencing was performed at

the CCHMC DNA Sequencing and Genotyping Core. For each sequenced read, five end bases were trimmed. Reads were further trimmed if more than 10% of bases with a quality score < 20. The remaining reads were aligned against human genome build hg19. Alignments with MAPping Quality (MAPQ) quality score ≥ 10 were included in the peak calling. Peaks were called using MACS2 callpeak v2.1.0 (Zhang *et al*, 2008; https://github.com/taoliu/MACS/) with the following parameters "–gsize 2451960000 –bw = 300 –ratio 1.0 –slocal 1000 –llocal 10000 –keep-dup 1 –bdg –qvalue 0.05″. The DNA sequences of peaks were retrieved using RSAT "fetch sequences from UCSC" tool (http://rsat.sb-roscoff.fr/). The genome distribution of ChIP-seq peaks was analyzed using CEAS v1.0.0 (Shin *et al*, 2009; http://liulab.dfci.harvard.edu/CEAS/) in Cistrome (http://cistrome.org/ap/). Motif analysis was performed using RSAT peak-motifs (Thomas-Chollier *et al*, 2012a,b; http://floresta.eead.csic.es/rsat/peak-motifs_form.cgi). For ChIP-qPCR, DNA was quantified after ChIP using SYBR Green (cat# 4385612, Thermo Fisher Scientific) on StepOnePlus Real-Time PCR System (Thermo Fisher Scientific) with primers (forward: 5′-CCCCTGGTAAAGTCGGGAAG-3′; reverse: 5′-GTGGCAGAATCAGGAAGCCT-3′ for the *MUC5AC* locus, forward: 5′-GCTGTGTCCCTTTCCTTCCT-3′; reverse: 5′-GGCACACAGTGACACCAAAC-3′ for the *MUC5B* locus and forward: 5′-GTGCCAGATAGTGCGGGTAT-3′; reverse: 5′-CACAGCAGGCCCAATTAGAT-3′ for the *ACTB* locus).

## CRISPR/Cas9 experiments

CRISPR/Cas9 vectors targeting the *MUC5AC* and *MUC5B* loci were made by inserting single-guide RNAs into pCas-Guide-EF1a-GFP vector (Origene Rockville, MD). The vectors were transiently transfected into A549 cells using Lipofectamine 3000 (Thermo Fisher Scientific). GFP-positive cells were sorted and plated at one cell per well on 96-well plates by the Research Flow Cytometry Core at CCHMC. DNA was extracted from each cell. The deletion of the *MUC5AC* locus was confirmed by PCR using primers (forward: 5′-CCCGGGAAGATGAAGATGGA-3′; reverse: 5′-TGTTCTGCCATTTCTGCCAC-3′). The deletion of the *MUC5B* locus was confirmed by PCR using primers (forward: 5′-GGCTCTGAGCAGACCAAGAG-3′; reverse: 5′-AGCCTAGTGTCTCCCCTGCT-3′). Deletion of the locus was further confirmed by Sanger sequencing at the CCHMC DNA Sequencing and Genotyping Core using the same PCR primers.

## Data availability

The RNA-seq and ChIP-seq data from this publication have been deposited in Gene Expression Omnibus (GEO) database (http://www.ncbi.nlm.nih.gov/geo/) under accession code GSE86959. See Dataset EV8 for the summary of the datasets.

## Statistical analysis

Sample sizes were designed to give statistical power. For the study using human specimens, a minimum of six patient specimens was used for RNA sample analysis. A minimum of 14 patient specimens was used for histological analysis. Assessment of histology was performed by three investigators to avoid subjective bias (K.T., I.M.F-B., and Y.M.). For the mouse study while minimizing animal use, a minimum of eight mice in each genotype was enrolled for the

## The paper explained

### Problem

Invasive mucinous adenocarcinoma of the lung (IMA) is pathologically well defined as a lung tumor with goblet cell morphology containing excessive intracytoplasmic mucins. However, the molecular pathway driving IMA is not well understood, thereby few therapeutics have been identified to treat IMA. Recent advances in mouse models have identified genes involved in causing IMA in mice; however, their relevance with human IMA is not known.

### Results

Using gene expression profiles from the mouse models and human IMA specimens, we determined a gene signature of 143 genes common in mouse and human IMA. The signature includes therapeutic targets, including an immune checkpoint *VTCN1/B7-H4*. Key transcription factors FOXA3 and SPDEF in the signature were sufficient to induce mucinous lung tumors in the presence of a *KRAS* mutation. ChIP-seq analysis determined that SPDEF bound to the non-coding upstream regions of the mucin genes *MUC5AC* and *MUC5B* that are highly expressed in human IMA. Deletion of the SPDEF binding regions by CRISPR/Cas9 significantly reduced the expression of the mucin genes.

### Impact

Human IMA is a lung tumor that is so far only pathologically defined. This study has determined a gene signature to define human IMA at the molecular level in addition to the pathological definition. The signature includes novel genes expressed in human IMA, which were not previously reported. The signature also includes therapeutic targets and genes driving IMA. The novel mouse models that we have developed will be useful as preclinical models for testing therapeutic drugs. The identification of the gene regulatory mechanisms inducing mucin genes will also contribute to the understanding of other mucin-producing diseases such as breast cancer and idiopathic pulmonary fibrosis (IPF).

survival study. For the histological study, a minimum of five mice in each genotype that developed lung tumors was enrolled and a minimum of three mice in each genotype that did not develop lung tumors was enrolled. All of the double-transgenic mice were littermates from the triple transgenic mouse development and co-housed with the triple transgenic mice. See Dataset EV6 for further mouse information. Biological triplicates were used for the RNA-seq study using the A549 cell line. Biological duplicates were used for the ChIP-seq study of each transcription factor. Statistical differences were determined using two-tailed paired Student's *t*-test (RNA-seq gene expression study using human specimens and ChIP-qPCR), Mann–Whitney test (Taqman gene expression qPCR study using human specimens), two-tailed unpaired Student's *t*-test (Taqman gene expression qPCR study using cell lines), Wald test in DESeq2 (Love *et al*, 2014; RNA-seq differential expression study using the A549 cell line), statistical test in MACS2 (Zhang *et al*, 2008; ChIP-seq peak calling study), or log-rank (Mantel–Cox) test (Kaplan–Meier survival mouse study by Prism 6 [GraphPad Software, La Jolla, CA]). The difference between two groups was considered significant when the *P*-value was < 0.05 and Benjamini–Hochberg-adjusted *P*-value < 0.05 when multiple comparisons were performed.

**Expanded View** for this article is available online.

## Acknowledgements

We thank J. Whitsett, F. Sladek, M. Durbin, I. Mitsui, H. Hao, P. Dexheimer, D. Fletcher, K. Dillehay McKillip, M. Fallenbüchel, G. Chen, Y-C. Hu, H. Miyoshi, F. Accornero, T. Suzuki and A. Perl for materials, technical support, and discussions. This work was supported by the American Lung Association (RG309608), Trustee Award Grant, CF-RDP Pilot & Feasibility Grant, Cincinnati Children's Hospital Medical Center (Y.M.), the Rotary Foundation Global Grant (K.T.), and the Ministry of Education, Science and Culture, Japan (T.T., T.N., Y.N., and T.F.).

## Author contributions

YM designed experiments. MG and YX designed bioinformatics analytic approaches. MG and RK performed bioinformatical analysis. KT, IMF-B, and YM performed experiments. MMe, TM, TF, TT, AW, TN, YN, and MMa provided human specimens. All authors contributed to experimental interpretation and manuscript writing.

## Conflict of interest

The authors declare that they have no conflict of interest.

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
