## [Review Process File · EMBO Molecular Medicine]

Gene signature driving invasive mucinous adenocarcinoma of the lung

Minzhe Guo, Koichi Tomoshige, Michael Meister, Thomas Muley, Takuya Fukazawa, Tomoshi Tsuchiya, Rebekah Karns, Arne Warth, Iris M. Fink-Baldauf, Takeshi Nagayasu, Yoshio Naomoto, Yan Xu¹, Marcus A. Mall and Yutaka Maeda

Corresponding author: Yutaka Maeda, Cincinnati Children's Hospital Medical Center

Review timeline:	Submission date:	16 June 2016
	Editorial Decision:	21 July 2016
	Revision received:	11 October 2016
	Editorial Decision:	04 November 2016
	Revision received:	08 January 2017
	Editorial Decision:	20 January 2017
	Revision received:	24 January 2017
	Accepted:	26 January 2017

Transaction Report:

Editor: Roberto Buccione

1st Editorial Decision

21 July 2016

Thank you for the submission of your manuscript to EMBO Molecular Medicine

We have now heard back from the three Reviewers whom we asked to evaluate your manuscript.

Although the reviewers find the manuscript to be of significant interest and generally well thought out, they do raise important concerns. I will not dwell into much detail, but I would like to highlight the main points.

The three reviewers find the manuscript to be of significant interest and generally well thought out. However, they also point to several important (and partially overlapping) issues. These include the generally poor presentation of the RNA-seq and ChIP-seq data, lack of access to the full dataset, various internal inconsistencies, poor characterisation of the TG mouse models, insufficient patient data, insufficient mechanistic insight and others. Some of these reflect my own my own reservations when making an initial editorial decision to send out for peer-review.

In conclusion, while publication of the paper cannot be considered at this stage, given the potential interest of your findings and after internal discussion, we have decided to give you the opportunity to address the criticisms. Please consider that the concerns raised are of great importance for us as they impinge on the overall quality and robustness of experimental support for the main

conclusions.

We are thus prepared to consider a substantially revised submission, with the understanding that the reviewers' concerns must be addressed with additional experimental data where appropriate and that acceptance of the manuscript will entail a second round of review.

EMBO Molecular Medicine now requires a complete author checklist (<http://embomolmed.embopress.org/authorguide#editorial3>) to be submitted with all revised manuscripts. Provision of the author checklist is mandatory at revision stage; The checklist is designed to enhance and standardize reporting of key information in research papers and to support reanalysis and repetition of experiments by the community. The list covers key information for figure panels and captions and focuses on statistics, the reporting of reagents, animal models and human subject-derived data, as well as guidance to optimise data accessibility.

As you know, EMBO Molecular Medicine has a "scooping protection" policy, whereby similar findings that are published by others during review or revision are not a criterion for rejection. However, I do ask you to get in touch with us after three months if you have not completed your revision, to update us on the status. Please also contact us as soon as possible if similar work is published elsewhere.

Please note that we now mandate that all corresponding authors list an ORCID digital identifier. You may do so through our web platform upon submission and the procedure takes <90 seconds to complete. We also encourage co-authors to supply an ORCID identifier, which will be linked to their name for unambiguous name identification.

Since the required revision in this case appears to require a significant amount of time, additional work and experimentation and might be technically challenging, I would therefore understand if you chose to rather seek publication elsewhere at this stage. Should you do so, we would welcome a message to this effect. Please note that it is EMBO Molecular Medicine policy to allow a single round of revision only and that, therefore, acceptance or rejection of the manuscript will depend on the completeness of your responses included in the next, final version of the manuscript.

I look forward to seeing a revised form of your manuscript as soon as possible.

***** Reviewer's comments *****

Referee #1 (Remarks):

Guo et al. reported in this manuscript the identification of a gene signature for human invasive mucinous adenocarcinoma of the lung (IMA), a subtype of lung adenocarcinoma (LUAD). This gene signature consisted of 143 genes that were higher expressed in human IMA and that were identified by comparing previously published gene expression analysis in murine models for IMA (Maeda et al., 2012; Snyder et al., 2013) and RNA-Seq based expression analysis in six human IMA samples in the current manuscript. The authors also found that the gene signature was enriched in mucin-producing cancers as gastrointestinal, pancreatic and breast cancer. Importantly, the authors detected high expression of VTCN1 in human IMA. VTCN1 is considered as an immune checkpoint protein since it is expected to be an antibody-mediated therapeutic target against breast cancer (Smyth et al., 2016; Leong et al., 2015), thereby suggesting the possibility for a similar therapeutic approach within the context of IMA. Further, the authors present *in vitro* and *in vivo* data supporting a molecular model for the formation of mucinous lung tumors, in which NKX2-1 acts as an "anti-mucous" transcription factor whereas FOXA3, SPDEF and HNF4A act as "pro-mucous" transcription factors within the context of active KRAS signaling. Based on the arguments mentioned above, this work constitutes a relevant contribution to the field of lung cancer with a significant translational potential, not only for the diagnosis but also for the development of therapeutic approaches against IMA. However, before recommending the manuscript for publication as Research Article at EMBO Mol Med, I would like to suggest several points that have to be improved:

Major comments:

1. The authors have to provide access to the RNA-Seq and ChIP-Seq data presented in the manuscript as stated in the authors guidelines from EMBO Mol Med. I was unable to assess the quality of these data since I did not have access to the raw data. For example, the authors can submit all the sequencing data included in the manuscript to NCBI GenBank (<http://www.ncbi.nlm.nih.gov/genbank/>), and generate a reviewer link in order that we (the Reviewers) have anonym and "read-only" access to the raw data.

In addition, in order to facilitate the quality assessment of the RNA-Seq and ChIP-Seq data, the authors should include a table containing columns with number of raw reads, number of trimmed reads, mapped reads and percentage of mapped reads.

2. The clinical relevant information of the patients used for the RNA-Seq was not provided. The authors should provide a table containing for example, age, gender, ethnic group, smoking history, treatment of the patient at the time point of sample collection and other information that might be relevant for further use of the RNA-Seq data by other researchers in the future.

3. The authors have to describe in the Material and Methods section how the heat maps presented in different figures of the manuscript were generated. For example, which software and which parameters were used for the generation of the heat maps.

4. The results obtained by RNA-Seq based expression analysis in human IMA samples should be confirmed, at least partially, by a second method. For example, the expression of MUC5AC, HNF4A, VTCN1, MUC5B, FOXA3, SPDEF, MUC3, PD-L1 and NKX2-1 should be analyzed by qRT-PCR in at least two cases each of human IMA and Normal Lung. I selected these genes to be confirmed because they are crucial in the model that the authors present at the end of the manuscript.

5. The authors should show scale bars in all the figure panels in which they presented pictures of immunohistochemistry (IHC).

6. In the Figure 2C, the authors have analyzed the Top200 VTCN1 and PD-L1 correlated genes with the human IMA. For the Top 200, they have used RNA-Seq Data (cell lines) from Klijn et al. 2015 and the TCGA LUAD 2014 data. I wonder why the two tables presented contain different genes and why SPDEF and MUC5B can only be found in the RNA Seq of Klijn et al. 2015 VTCN1 Top200. The authors should explain why the two lists of genes presented are different. More interesting, the authors should state how many of these genes are present in the gene signature of 143 genes identified in the Figure 1.

7. In the Figures 2E (using NSCLC cell lines) and 2G (using LUAD from the TCGA), there is a relatively strong positive correlation between the expression of NKX2-1 and SPDEF, what seems to be contradictory to the model suggested by the authors in Figure 5D. The authors should explain these observations.

8. The authors have found that VTCN1, but not PD-L1, is an immune checkpoint protein that is expressed in human IMA (Figure 1A, among others), thereby suggesting its potential use as a therapeutic target. The authors have also shown a positive correlation of FOXA3 and SPDEF with VTCN1 (Figures 2E and 2G). Furthermore, the authors demonstrated that the expression of Foxa3 and Spdef induces mucin expression (Figure 5A). Following this line of reasoning, instead of, or at least in addition to showing that PD-L1 is not present in human IMA due to a downregulation of NKX2-1 (Figure 3), it would be more beneficial for the strength of this paper, if the authors would focus on VTCN1. I suggest analyzing by ChIP the VTCN1 locus after immunoprecipitation of FOXA3 and/or SPDEF. Remarkably, the authors have presented ChIP-Seq results on the MUC5AC (Figure 6A) and MUC5B (Figure 7A) loci after immunoprecipitation of SPDEF in A549 and MCF7 cells. The authors should check in these ChIP-Seq data whether SPDEF binds to VTCN1. In addition, the authors should perform similar type of experiments as presented in the Figure 3, but demonstrating the regulation of VTCN1 by SPDEF and/or FOXA3.

9. In Figure 4A, the authors selected FOXA3 and SPDEF as their top 2 candidates to analyze their levels in human IMA (Figure 4B) and their functional role in murine IMA using transgenic mouse

models (Figures 4C-D). The authors perform IHC on human IMA using FOXA3-specific antibodies and Alcian Blue Staining (Figure 4B). However, they do not show the staining on sections of adjacent control samples. In addition the IHC for SPDEF is missing. The authors should improve these two points in this figure panel. Further, I suggest incorporating pie charts that correlates the expression of VTCN1 in FOXA3 positive and negative human IMAs as shown in Figure 3F for the correlation between PD-L1 and NKX2-1.

10. Related to the transgenic mice models (Figures 4C-D), the authors claim that Tet-inducible expression of Foxa3 or Spdef in Scgb1a1-positive cells in the absence of KRASG12D produced airway goblet cells. Further, inducible expression of KrasG12D in Scgb1a1-positive cells produced not mucinous lung tumors. Only the expression of KrasG12D in combination with Foxa3 or Spdef in Scgb1a1-positive cells produced mucinous lung tumors. In general, the authors should show these pictures with a higher magnification.

Further, it looks like that the induction of Foxa3 in Scgb1a1-positive cells was not efficient as shown by the FOXA3 staining (Figure 4C). The authors should provide a different representative figure.

It looks like that induction of KRASG12D in Scgb1a1-positive cells increased the levels of SPDEF (Figure 4D). If this is the case, the author should explain why is KRAS12D not able to produce mucinous tumors since MUC5AC and MUC5B are direct SPDEF targets (as shown in the Figures 6 and 7).

11. In A549 cells, the expression of Spdef increased the levels of FOXA3 and expression of Foxa3 increased the levels of SPDEF (Figure 5A, left). Further, transfection of Spdef or Foxa3 increased the levels of MUC5AC and MUC5B mRNAs (Figure 5A, right). Taking these data together, it will be interesting to investigate whether Spdef requires Foxa3 and vice versa for their effects on MUC5AC and MUC5B expression. For example, one could analyze the expression of MUC5AC and MUC5B after Spdef transfection and siRNA mediated FOXA3 loss-of-function. A similar experimental setting could be performed after Foxa3 transfection and SPDEF loss-of-function.

12. The ChIP-Seq results presented in the Figures 3A, 6A and 7A seem to be robust (several repetitions). However, the authors should present a genome browser picture of these ChIP-Seq data including the input of at least one repetition. It is well known that sometimes there are enriched areas in which transcriptions factors seem to bind but they are not called as peak due to a high background in the negative control or in the input. A snapshot of the UCSC genome browser or IGV would help to improve the presentation of the ChIP-Seq data. The authors should check this link to obtain detailed information on how to proceed (<http://homer.salk.edu/homer/ngs/tagDir.html>).

Minor comments:

13. Page 9: Figure 4G and H should be Figure 3G and H

14. Figure 1A, What does the grouping of the cases mean? Labeling of the color code is missing.

15. Figure 1D, labeling of color code is missing

16. Figure 2A, scale bar is missing.

17. Figure 2D-G, color code of arrows is missing

18. Figure 3C, what does CST, Spring, Sino mean?

19. Figure 4B, scale bar is missing.

20. Figure 5, the lettering of the panels in the figure do not match to the description in the legend, 1C is described as 1B, figure 1B is 1D, and 1D is 1C.

21. Figure 5C left panel, what does the star mean?

22. The loading controls of the WB are over-exposed. They authors should select less exposed bands

in order to show equal loading. Further, the authors wrote "Actin" as IB, they should be labeled as ACTA1 if used.

23. TaqMan Gene Expression, if GAPDH was used as loading control, it should be stated like this.

24. Time of dox administration should be explained in the experimental mice part.

25. The primers used for the genotyping of the MUC5b locus should be checked again, since Primer-Blast cannot find its appropriate binding sites.

26. Author's contribution: All authors contributed "to" experimental interpretation...

I hope that my suggestions contribute to an increase in the quality of the manuscript. I wish success in the revision of the manuscript.

Referee #2 (Comments on Novelty/Model System):

The authors use human cell lines, human tissue and animal models. Appropriate permission was obtained for the use of animals and human tissues.

Referee #2 (Remarks):

This study by Guo et al uses molecular profiling to determine genes implicated in invasive mucinous adenocarcinoma and identified two key regulators; FOXA3 and SPDEF. The importance of FOXA3 and SPDEF in mucinous lung tumors was confirmed in a transgenic mouse model and their ability to regulate MIC5AC and MUC5B confirmed using ChIP-seq and CRISPR targeting. This manuscript also identified VTCN1, an immune checkpoint gene, as a potential therapeutic target for invasive mucinous lung adenocarcinoma. While this study provides novel information I do have a number of concerns which are listed below.

1. While some aspects of this manuscript are explored in detail, others are only covered at a very superficial level. For example, the studies investigating the role of FOXA3 and SPDEF in mucinous lung cancer are very detailed (gene overexpression and promoter studies) while the work describing the transgenic animal model is very limited. Although extremely interesting, a more detailed characterization of the transgenic mouse model including SPDEF and FOXA3 expression levels +/- Dox over time, histological evaluation of the lungs at various time points, more detailed description of tumor onset, incidence and progression (or references to manuscripts previously published on this model) are required before this work should be published. Moreover, reporting in the discussion that 2 out of 11 transgenic mice in dox treated group died at 4 months but not providing detailed information on the reason(s) for these deaths, is not really useful.

2. Some discussion as to why the two IMA mouse models have such a different number of differentially expressed genes and why the authors focused on a 143 gene signature rather than a signature based on the 25 genes found in both mouse models and the human data should be provided. It would also have been useful to compare the gene expression profile of the 6 pairs of normal and IMA tumors used in this study to the 9 IMA pairs from TCGA. Where all the genes in the 143 gene signature found in both sets of samples?

3. In Figure 2E&G, the correlations between genes is quite different in the cell lines compare to the lung tumor tissue. The significance of this should be discussed

4. I am having a difficult time understanding Fig 2F. According to the legend this is hierarchical clustering of IMA-related genes and cell type markers. If the genes used are truly markers of IMA, how come they do such a poor job of clustering the IMA samples (indicated by red bars)?

5. The data presented in Fig 3F &H does not seem that useful. I understand that the authors are trying to show that mucinous tumors rarely express PD-L1 but the percentage of non-mucinous tumors expressing PD-L1 is also quite low. So I am not sure how clinically relevant this information

is. Although I am not an expert in this field, my understanding was that PD-L1 inhibitors have not really shown dramatic results in NSCLC patients.

Some Minor Points

1. It would be useful if the authors used arrows or lines to clearly indicate which row belongs to each gene listed in Fig 1A
2. Stomach cancer and colorectal cancer cell enrichment scores have been reversed in the first line of page 7...colorectal cancer has enrichment score of 0.81 according to Fig 1E

Referee #3 (Comments on Novelty/Model System):

The manuscript is well written. I have included some comments to the authors below that I hope will make the paper better.

I find no reason to suspect ethical issues will result from this paper.

Referee #3 (Remarks):

The authors have attempted to elucidate the gene signature of Invasive Mucinous Adenocarcinoma (IMA), a rare type of lung adenocarcinoma (LUAD), and as of yet understudied.

This signature turned up among other things, the potential gene target VTCN1.

This manuscript appears well polished, well thought out and well written.

Major Critiques:

VTCN1 is proposed as a therapeutic target for IMA but is not found to be expressed at the protein level in all IMAs. How many conditions were tested for optimized VTCN1 staining? How many different clones of this antibody were tested? In the event it is only one, why? The reviewer would question, how do you propose to use this as a therapeutic target if it is not present? This leads me to question the robustness of the generated signature. Please discuss.

Related but in general, more important than catalogue numbers for your antibodies are the clone names. Please include. Catalogue numbers are of course useful, but to be truly reproducible, the same clone (for monoclonals) must be used between studies as well. Polyclonals are best avoided and their use appears limited in this manuscript.

Minor Critiques:

Data would be easier to interpret if the scales on 2D and 2F were the same; preferably 2D would be altered to fit scaling with 2F

Reference to Fig4G,H in the manuscript. There is no Fig4G and H. Do you mean Fig3G,H? Please revise. Reread your submissions carefully (including this one) to correct errors overlooked in manuscript writing and layout.

Minor Critiques:

Mucous production by tumors has been speculated to contribute to the invasive nature of these types of cancers. In order to foster ideas for the cancer research community branching out from this very rare cancer type, please discuss how this is relevant to your observations in pancreatic and breast tumors.

IMA is known to be "defective in NKX2-1 expression" or lacking in NKX2-1 expression? Which is more accurate? Revise wording if necessary.

1st Revision - authors' response

11 October 2016

Referee #1 (Remarks):

Guo et al. reported in this manuscript the identification of a gene signature for human invasive mucinous adenocarcinoma of the lung (IMA), a subtype of lung adenocarcinoma (LUAD). This gene signature consisted of 143 genes that were higher expressed in human IMA and that were identified by comparing previously published gene expression analysis in murine models for IMA (Maeda et

al., 2012; Snyder et al., 2013) and RNA-Seq based expression analysis in six human IMA samples in the current manuscript. The authors also found that the gene signature was enriched in mucin-producing cancers as gastrointestinal, pancreatic and breast cancer. Importantly, the authors detected high expression of VTCN1 in human IMA. VTCN1 is considered as an immune checkpoint protein since it is expected to be an antibody-mediated therapeutic target against breast cancer (Smyth et al., 2016; Leong et al., 2015), thereby suggesting the possibility for a similar therapeutic approach within the context of IMA. Further, the authors present in vitro and in vivo data supporting a molecular model for the formation of mucinous lung tumors, in which NKX2-1 acts as an "anti-mucous" transcription factor whereas FOXA3, SPDEF and HNF4A act as "pro-mucous" transcription factors within the context of active KRAS signaling. Based on the arguments mentioned above, this work constitutes a relevant contribution to the field of lung cancer with a significant translational potential, not only for the diagnosis but also for the development of therapeutic approaches against IMA. However, before recommending the manuscript for publication as Research Article at EMBO Mol Med, I would like to suggest several points that have to be improved:

Major comments:

Comment 1: The authors have to provide access to the RNA-Seq and ChIP-Seq data presented in the manuscript as stated in the authors guidelines from EMBO Mol Med. I was unable to assess the quality of these data since I did not have access to the raw data. For example, the authors can submit all the sequencing data included in the manuscript to NCBI GenBank (https://urldefense.proofpoint.com/v2/url?u=http-3A__www.ncbi.nlm.nih.gov_genbank_&d=CwIDaQ&c=P0c35rBvln7D8BNx7kSJTg&r=pb95mIrbvTNnEFSt4E3HzbJ6CkSaGYfETXkAI9wv778&m=NC3hRiXfl5RVMWWzVBsN_1v5kIaPMTilp3ewTy0v6wY&s=5p0LR0iPyB20K4qxt_0AOtk-IUNpc-RJmiV-Ang78g&e=), and generate a reviewer link in order that we (the Reviewers) have anonym and "read-only" access to the raw data. In addition, in order to facilitate the quality assessment of the RNA-Seq and ChIP-Seq data, the authors should include a table containing columns with number of raw reads, number of trimmed reads, mapped reads and percentage of mapped reads.

Response 1: According to the guidelines, we have now submitted the raw data of the RNA-seq and ChIP-seq data to the NCBI domain. Access to the data can be obtained by following this reviewer link:

<https://www.ncbi.nlm.nih.gov/geo/query/acc.cgi?token=kxqzuwcydvsprkn&acc=GSE86959>.

We also added a new Appendix Table S8 containing columns with the number of raw reads, number of trimmed reads, mapped reads and percentage of mapped reads.

Comment 2: The clinical relevant information of the patients used for the RNA-Seq was not provided. The authors should provide a table containing for example, age, gender, ethnic group, smoking history, treatment of the patient at the time point of sample collection and other information that might be relevant for further use of the RNA-Seq data by other researchers in the future.

Response 2: We added the patient information in the Appendix Table S1.

Comment 3: The authors have to describe in the Material and Methods section how the heat maps presented in different figures of the manuscript were generated. For example, which software and which parameters were used for the generation of the heat maps.

Response 3: In the revised manuscript, we described the software and parameters to generate the heat maps in the corresponding analysis sections in Materials and Methods.

Comment 4: The results obtained by RNA-Seq based expression analysis in human IMA samples should be confirmed, at least partially, by a second method. For example, the expression of MUC5AC, HNF4A, VTCN1, MUC5B, FOXA3, SPDEF, MUC3, PD-L1 and NKX2-1 should be analyzed by qRT-PCR in at least two cases each of human IMA and Normal Lung. I selected these genes to be confirmed because they are crucial in the model that the authors present at the end of the manuscript.

Response 4: We performed Taqman gene expression qPCR analysis using RNAs from 7 IMA and 6 normal lung specimens available in the Biobank, from which 2 IMA and 2 normal lung tissues were used for the RNA-seq analysis to generate Fig 1A. Please see the Fig EV1 for the qPCR data and the Appendix Table S1 for the patients' information. The expression of *MUC5AC*, *HNF4A*, *VTCN1*, *MUC5B*, *FOXA3*, *SPDEF* and *MUC3* was significantly induced in IMA compared to normal lung tissues, consistent with the RNAseq data. *PD-L1* and *NKX2-1* were reduced (but not induced). The reduction of *PD-L1* was not significant by the Taqman gene expression qPCR data (Fig EV1), inconsistent with the RNA-seq data (Fig 1A); nevertheless, both data indicates that *PD-L1* is not induced in IMA compared to normal lung tissues. We clearly stated that *PD-L1* was not induced in IMA in the text. Consistent with the RNA-seq data (Fig 1A), these results indicate that *VTCN1* (B7-H4) is a better immune checkpoint than *PD-L1* (B7-H1) to target IMA. Reduction of *NKX2-1* in IMA at the protein level is well established by immunohistochemistry (Travis et al, 2011); however, *NKX2-1* was not significantly reduced in IMA at the mRNA level compared to normal lung tissue in the RNA-seq (Fig 1A) and the Taqman gene expression qPCR analyses (Fig EV1; $P=0.051$). The mechanism by which *NKX2-1* is reduced in IMA at the mRNA or protein level is not well understood. Further study is required to elucidate the mechanism.

Comment 5: The authors should show scale bars in all the figure panels in which they presented pictures of immunohistochemistry (IHC).

Response 5: We added scale bars in all the figure panels in the revised manuscript.

Comment 6: In the Figure 2C, the authors have analyzed the Top200 VTCN1 and PD-L1 correlated genes with the human IMA. For the Top 200, they have used RNA-Seq Data (cell lines) from Klijn et al. 2015 and the TCGA LUAD 2014 data. I wonder why the two tables presented contain different genes and why SPDEF and MUC5B can only be found in the RNA Seq of Klijn et al. 2015 VTCN1 Top200. The authors should explain why the two lists of genes presented are different. More interesting, the authors should state how many of these genes are present in the gene signature of 143 genes identified in the Figure 1.

Response 6: In the revised manuscript, we reanalyzed the Pearson's correlations of gene expression in two datasets using the transformed gene expression data utilized in the original articles (Klijn et al, 2015; Cancer Genome Atlas Research Network, 2014) rather than using expression values quantile-normalized by ourselves in our previous manuscript. For the 105 NSCLC cell line dataset, we downloaded the variancestabilizing transformation (VST) of gene expression generated by DESeq2 (Love et al, 2014) from the original paper (Klijn et al, 2015). For the 230 TCGA LUAD dataset, we downloaded the normalized expression data, which were quantified using RSEM (Li and Dewey, 2011), normalized within-sample to a fixed upper quartile with zero values imputed by the overall minimum value and log2 transformed from the original paper (Cancer Genome Atlas Research Network, 2014). Both transformed datasets were in log2 scale, normalized with respect to library size and utilized in the original articles for downstream analysis, such as clustering. For the 230 TCGA LUAD dataset, the reanalyzed *VTCN1* or *PD-L1* highly correlated genes in the revised manuscript are exactly the same as the ones in our previous manuscript. For the 105 NSCLC cell line dataset, both the numbers of *VTCN1* and *PD-L1* highly correlated IMA-expressing genes were increased; nevertheless, the number associated with *VTCN1* ($n=38$) is still significantly higher (one tailed Fisher's exact test: $P\text{-value}=4.148\text{e-}07$) than the number associated with *PD-L1* ($n=7$). Now the two tables contain 11 common genes (*ALDH3B2*, *ELF3*, *EPN3*, *GRHL1*, *MUC20*, *OVOL1*, *OVOL2*, *PLEKHG6*, *PRSS22*, *PVRL4* and *STI4*) as *VTCN1* Top 200 correlated genes (Fig 3C). As pointed by the reviewer, they also contain different genes. We used the two datasets to take advantage of available data but differences in the way that the two datasets were generated gave different results. The 230 TCGA LUAD dataset is from an intact lung tumor; however, it is composed of the combined gene expression profile of not only lung tumor cells but also tumor-associated fibroblasts, endothelial and immune cells, which does not provide a gene expression profile of the solo lung tumor cells. The 105 NSCLC cell line dataset is composed of the gene expression profile of only tumor cells; however, each cell line is derived from an isolated tumor cell after multiple passages on plastic dishes, which may not represent lung tumor cells in an intact lung tumor environment. This might be the reason why there are different genes in the two datasets. We discussed the strengths and the limitations of these two datasets. In the revised figure

(Fig 3C), we also highlighted genes in red, which are in the Mucinous Lung Tumor Signature of 143 genes.

Comment 7: In the Figures 2E (using NSCLC cell lines) and 2G (using LUAD from the TCGA), there is a relatively strong positive correlation between the expression of NKX2-1 and SPDEF, what seems to be contradictory to the model suggested by the authors in Figure 5D. The authors should explain these observations.

Response 7: As discussed in Response 6, in the revised manuscript, we reanalyzed the Pearson's correlations of gene expression in two datasets using the data transformations utilized in the original articles (Klijn et al, 2015; Cancer Genome Atlas Research Network, 2014), which are in log₂ scale and normalized with respect to library size. In the revised manuscript, Fig 3E (previously Fig 2E) and Fig 3G (previously Fig 2G) were re-generated using the updated correlation values. We looked at the correlations between *NKX2-1* and *SPDEF* expression in both datasets. There is no strong positive correlation ($\rho=0.036$) between *NKX2-1* and *SPDEF* expression in the 230 TCGA LUAD dataset. The correlation between *NKX2-1* and *SPDEF* expression in the 105 NSCLC cell line dataset is positive ($\rho=0.311$). However, when we exclude 41 cell lines that do not express *NKX2-1* and *SPDEF* (RPKM \leq 1) from the calculation, there is no positive correlation ($\rho=-0.042$) between *NKX2-1* and *SPDEF* expression in the remaining 64 cell lines, suggesting that the positive correlation was seen due to the large number of the cell lines that lack the expression of both *NKX2-1* and *SPDEF*.

Comment 8: The authors have found that VTCN1, but not PD-L1, is an immune checkpoint protein that is expressed in human IMA (Figure 1A, among others), thereby suggesting its potential use as a therapeutic target. The authors have also shown a positive correlation of FOXA3 and SPDEF with VTCN1 (Figures 2E and 2G). Furthermore, the authors demonstrated that the expression of Foxa3 and Spdef induces mucin expression (Figure 5A). Following this line of reasoning, instead of, or at least in addition to showing that PD-L1 is not present in human IMA due to a downregulation of NKX2-1 (Figure 3), it would be more beneficial for the strength of this paper, if the authors would focus on VTCN1. I suggest analyzing by ChIP the VTCN1 locus after immunoprecipitation of FOXA3 and/or SPDEF. Remarkably, the authors have presented ChIP-Seq results on the MUC5AC (Figure 6A) and MUC5B (Figure 7A) loci after immunoprecipitation of SPDEF in A549 and MCF7 cells. The authors should check in these ChIP-Seq data whether SPDEF binds to VTCN1. In addition, the authors should perform similar type of experiments as presented in the Figure 3, but demonstrating the regulation of VTCN1 by SPDEF and/or FOXA3.

Response 8: Our ChIP-seq data (A549 lung carcinoma cells) and that of others (MCF7 breast adenocarcinoma cells) indicate that SPDEF bound to the locus of *VTCN1* in MCF7 cells but not in A549 cells, suggesting a cell-dependent association of SPDEF with the locus of *VTCN1*. In addition, SPDEF did not induce *VTCN1* in A549 cells (Appendix Table S7). These results are now shown in Appendix Fig S3A in the revised manuscript. In the other ChIP-seq data, *NKX2-1* bound to the locus of *VTCN1* in A549 cells (Appendix Fig S3A), which suggests that *NKX2-1* might repress the expression of *VTCN1*. However, we were not able to see such repression by *NKX2-1* since A549 cells do not express the endogenous *VTCN1* (Fig 3D). The induction of *VTCN1* by *NKX2-1* was also not observed in A549 cells (Maeda et al, 2012). A549 cells may not be an appropriate cell line to assess the regulatory mechanism of *VTCN1* gene expression in LUAD, including IMA. Experiments using other lung cancer cell lines that express endogenous *VTCN1* (e.g., H1437, H2126, H292, H1781 etc.; see Fig 3D) may be suitable to understand the mechanism by which *VTCN1* is regulated in LUAD. In addition to the cell line issue, the expression of a transcription factor *ELF3* was highly correlated with *VTCN1* in both the 230 TCGA LUAD dataset (Cancer Genome Atlas Research Network, 2014) and the 105 NSCLC cell line dataset (Klijn et al, 2015), which is now shown in Fig 3C in the revised manuscript, suggesting that *ELF3* but not *SPDEF* may regulate the expression of *VTCN1*. Contrary to the mechanism by which *PD-L1* is regulated, the mechanism by which *VTCN1* is regulated is not well understood. We stated this limitation and potential regulatory mechanisms of *VTCN1* gene expression in the discussion section in the revised manuscript.

Comment 9: In Figure 4A, the authors selected FOXA3 and SPDEF as their top 2 candidates to analyze their levels in human IMA (Figure 4B) and their functional role in murine IMA using transgenic mouse models (Figures 4C-D). The authors perform IHC on human IMA using FOXA3-

specific antibodies and Alcian Blue Staining (Figure 4B). However, they do not show the staining on sections of adjacent control samples. In addition the IHC for SPDEF is missing. The authors should improve these two points in this figure panel. Further, I suggest incorporating pie charts that correlates the expression of VTCN1 in FOXA3 positive and negative human IMAs as shown in Figure 3F for the correlation between PD-L1 and NKX2-1.

Response 9: In the revised manuscript, we have now added the FOXA3 staining on sections of adjacent control samples (Fig 5B). From our experience (Chen et al, 2009), only antibody developed by Dr. Dennis Watson at University of South Carolina has been able to detect endogenous SPDEF expressed in human lung tissues. However, Dr. Watson does not have the antibody anymore, so we were not able to perform the IHC. As suggested, we included pie charts depicting the correlation of VTCN1 in FOXA3-positive or negative human IMA in Appendix Fig S3B in the revised manuscript. The majority of FOXA3-positive IMA expressed VTCN1; however, FOXA3-negative IMA also expressed VTCN1, suggesting that there might be additional regulators other than FOXA3 to induce VTCN1 (e.g., ELF3; see Response 8). We stated the results in the text as well.

Comment 10: Related to the transgenic mice models (Figures 4C-D), the authors claim that Tet-inducible expression of Foxa3 or Spdef in Scgb1a1-positive cells in the absence of KRASG12D produced airway goblet cells. Further, inducible expression of KrasG12D in Scgb1a1-positive cells produced not mucinous lung tumors. Only the expression of KrasG12D in combination with Foxa3 or Spdef in Scgb1a1-positive cells produced mucinous lung tumors. In general, the authors should show these pictures with a higher magnification. Further, it looks like that the induction of Foxa3 in Scgb1a1-positive cells was not efficient as shown by the FOXA3 staining (Figure 4C). The authors should provide a different representative figure. It looks like that induction of KRASG12D in Scgb1a1-positive cells increased the levels of SPDEF (Figure 4D). If this is the case, the author should explain why is KRAS12D not able to produce mucinous tumors since MUC5AC and MUC5B are direct SPDEF targets (as shown in the Figures 6 and 7).

Response 10: In the revised manuscript, we added pictures with a higher magnification. We also replaced the FOXA3 staining in the figure. For the revised manuscript, we used a different antibody produced from guinea pig, which was previously used in Chen et al, 2009, in order to detect the SPDEF staining in the mouse lung tissues. In the previous manuscript, we used an antibody from rabbit recently developed at our institute. We need to further characterize this rabbit antibody. We did not include the rabbit antibody made at our institute in the present manuscript and it is now clear in a higher magnification picture that SPDEF is expressed in the nucleus only in SPDEF-expressing mice but not in KRASG12D-expressing mice. We appreciated this careful observation by the reviewer.

Comment 11. In A549 cells, the expression of Spdef increased the levels of FOXA3 and expression of Foxa3 increased the levels of SPDEF (Figure 5A, left). Further, transfection of Spdef or Foxa3 increased the levels of MUC5AC and MUC5B mRNAs (Figure 5A, right). Taking these data together, it will be interesting to investigate whether Spdef requires Foxa3 and vice versa for their effects on MUC5AC and MUC5B expression. For example, one could analyze the expression of MUC5AC and MUC5B after Spdef transfection and siRNA mediated FOXA3 loss-of-function. A similar experimental setting could be performed after Foxa3 transfection and SPDEF loss-of-function.

Response 11: We performed the suggested experiment. We infected A549 cells with control or SPDEF-expressing lentivirus and transfected with control or FOXA3 siRNA. We extracted mRNA and assessed the expression of MUC5AC and MUC5B induced by SPDEF in the presence or absence of FOXA3. SPDEF was able to induce the expression of MUC5AC and MUC5B in the absence of FOXA3, indicating that SPDEF directly induces the mucous gene expression independently of FOXA3. The data is now shown in Appendix Fig S2 in the revised manuscript. Likewise, FOXA3 is able to induce the mucous gene expression in the absence of SPDEF, which was shown previously (Chen et al, 2014). We cited this paper in the text. These results indicate that SPDEF or FOXA3 independently induces the mucous gene expression.

Comment 12. The ChIP-Seq results presented in the Figures 3A, 6A and 7A seem to be robust (several repetitions). However, the authors should present a genome browser picture of these ChIP-Seq data including the input of at least one repetition. It is well known that sometimes there are

enriched areas in which transcriptions factors seem to bind but they are not called as peak due to a high background in the negative control or in the input. A snapshot of the UCSC genome browser or IGV would help to improve the presentation of the ChIP-Seq data. The authors should check this link to obtain detailed information on how to proceed (https://urldefense.proofpoint.com/v2/url?u=http-3A__homer.salk.edu_homer_ngo_tagDir.html&d=CwIDaQ&c=P0c35rBvln7D8BNx7kSJtg&r=pb95mIrbvTNnEFSt4E3HzbJ6CkSaGYfETXkAI9wv778&m=NC3hRiXfl5RVMWWzVBsN_1v5kIaPMTiIp3ewTy0v6wY&s=WTReHqtpjOvSccJLwUyFCZmZXxIl5iNexfBpJWk13D8&e=).

Response 12: In the revised manuscript, we added genome browser pictures of the ChIPseq data (see Fig EV5 and Appendix Fig S1). The peak calling was performed using MACS2 as described in Materials and Methods in the revised manuscript.

Minor comments:

Comment 13: Page 9: Figure 4G and H should be Figure 3G and H.

Response 13: We corrected them.

Comment 14: Figure 1A, What does the grouping of the cases mean? Labeling of the color code is missing.

Response 14: The case means the patient case. We stated that in the legend in the revised manuscript. We also labeled the color code.

Comment 15: Figure 1D, labeling of color code is missing

Response 15: We labeled the color code in the revised manuscript.

Comment 16: Figure 2A, scale bar is missing.

Response 16: We added the scale bar in the revised manuscript.

Comment 17: Figure 2D-G, color code of arrows is missing.

Response 17: We removed arrows and colored the genes in red or green. We indicated the description of the colors in the legends.

Comment 18: Figure 3C, what does CST, Spring, Sino mean?

Response 18: CST, Spring and Sino mean the companies that made the PD-L1 antibodies as described in the Materials and Methods. Since the cancer immunotherapy community is interested in which PD-L1 antibody works, we listed these three working antibodies to detect endogenous PD-L1. We also described that in the legend in the revised manuscript in addition to the description in the Materials and Methods.

Comment 19: Figure 4B, scale bar is missing.

Response 19: We added the scale bar in the revised manuscript.

Comment 20: Figure 5, the lettering of the panels in the figure do not match to the description in the legend, 1C is described as 1B, figure 1B is 1D, and 1D is 1C.

Response 20: We corrected them in the revised manuscript.

Comment 21: Figure 5C left panel, what does the star mean?

Response 21: The star means the endogenous SPDEF protein. We stated that in the legend in the revised manuscript.

Comment 22: The loading controls of the WB are over-exposed. They authors should select less exposed bands in order to show equal loading. Further, the authors wrote "Actin" as IB, they should be labeled as ACTA1 if used.

Response 22: We replaced the over-exposed ones to less exposed ones in the revised manuscript. We also changed Actin to ACTA1.

Comment 23: TaqMan Gene Expression, if GAPDH was used as loading control, it should be stated like this.

Response 23: We stated that in the legend in the revised manuscript.

Comment 24: Time of dox administration should be explained in the experimental mice part.

Response 24: We added a scheme describing the time of dox administration in addition to the strategy of the transgenic mouse development, which is in Fig EV3A.

Comment 25: The primers used for the genotyping of the MUC5b locus should be checked again, since Primer-Blast cannot find its appropriate binding sites.

Response 25: We corrected the primer sequence. We appreciated the careful review. In addition, we noticed that we wrote wrong numbers for the chromosome locations that were deleted in the *MUC5AC* locus, so we corrected that as well.

Comment 26: Author's contribution: All authors contributed "to" experimental interpretation...

Response 26: We corrected it.

Comment 27: I hope that my suggestions contribute to an increase in the quality of the manuscript. I wish success in the revision of the manuscript.

Response 27: We appreciate the reviewer's careful comments.

Referee #2 (Comments on Novelty/Model System):

The authors use human cell lines, human tissue and animal models. Appropriate permission was obtained for the use of animals and human tissues.

Referee #2 (Remarks):

This study by Guo et al uses molecular profiling to determine genes implicated in invasive mucinous adenocarcinoma and identified two key regulators; FOXA3 and SPDEF. The importance of FOXA3 and SPDEF in mucinous lung tumors was confirmed in a transgenic mouse model and their ability to regulate MIC5AC and MUC5B confirmed using ChIP-seq and CRISPR targeting. This manuscript also identified VTCN1, an immune checkpoint gene, as a potential therapeutic target for invasive mucinous lung adenocarcinoma. While this study provides novel information I do have a number of concerns which are listed below.

Comment 1: While some aspects of this manuscript are explored in detail, others are only covered at a very superficial level. For example, the studies investigating the role of FOXA3 and SPDEF in mucinous lung cancer are very detailed (gene overexpression and promoter studies) while the work describing the transgenic animal model is very limited. Although extremely interesting, a more detailed characterization of the transgenic mouse model including SPDEF and FOXA3 expression levels +/- Dox over time, histological evaluation of the lungs at various time points, more detailed description of tumor onset, incidence and progression (or references to manuscripts previously published on this model) are required before this work should be published. Moreover, reporting in the discussion that 2 out of 11 transgenic mice in dox treated group died at 4 months but not providing detailed information on the reason(s) for these deaths, is not really useful.

Response 1: The mouse models conditionally expressing KRASG12D, SPDEF or FOXA3 in lung epithelium were previously reported (Fisher et al, 2001; Park et al, 2007; Chen et al, 2014). The

tumor onset, incidence and progression of lung tumors in the KRASG12D double transgenic mouse are well established (Fisher et al, 2001). In the revised manuscript, we cited these papers in the text and added the strategy of the development of the triple transgenic mice using these transgenic mice along with a scheme of doxycycline (Dox) administration in Fig EV3A. As described in Fig EV3A, since these mouse models express KRASG12D, SPDEF and/or FOXA3 only in lung epithelium, the mortality in the models is most likely caused by tumors developed in the lung, which would result in lung dysfunction (e.g., airway obstruction by lung tumor and/or mucus) and/or cachexia (often seen in lung cancer mouse models; Maeda et al, 2012). However, the precise reasons for the death need further studies, including lung function test in the mouse models. In the revised manuscript, we included the Kaplan-Meier survival data of both the mouse models (KRASG12D/SPDEF or KRASG12D/FOXA3) in Fig EV3C, which indicates time of survival of the transgenic mice. As pointed out, it is ideal to take different time points (1, 2, 3...months) for each triple transgenic mouse; however, since lung tumorigenesis in the KRASG12D double transgenic mice is well established by Fisher et al, 2001, we selected a 4-month time point for the mouse group of KRASG12D/SPDEF and a 6-month time point for the mouse group of KRASG12D/FOXA3, based on time points when a portion of the mice in each group started dying (Fig EV3C). This experimental design also met our primary goal to determine whether FOXA3 or SPDEF in the presence of KRASG12D drives mucinous lung tumors *in vivo* while minimizing the use of the mice (we can normally obtain only one triple transgenic mouse from one litter). As suggested, in the revised manuscript, we performed the careful histological evaluation at the time points based on a paper reported by Sutherland et al, 2014 (cited in the revised manuscript). We appreciated this suggestion since we were able to differentiate the lung tumor phenotype of KRASG12D/SPDEF from the one of KRASG12D/FOXA3. Though both the KRASG12D/SPDEF mice and the KRASG12D/FOXA3 mice develop mucinous lung tumors, the KRASG12D/SPDEF mice developed malignant mucinous tumors (tubulopapillary-like carcinoma) while the KRASG12D/FOXA3 developed only benign mucinous tumors, which also implicates the difference in the survival of the two groups. We stated these observations in the text and included them in Fig EV3B in the revised manuscript. All of the mice information is described in Appendix Table S6.

Comment 2: Some discussion as to why the two IMA mouse models have such a different number of differentially expressed genes and why the authors focused on a 143 gene signature rather than a signature based on the 25 genes found in both mouse models and the human data should be provided. It would also have been useful to compare the gene expression profile of the 6 pairs of normal and IMA tumors used in this study to the 9 IMA pairs from TCGA. Where all the genes in the 143 gene signature found in both sets of samples?

Response 2: Our mouse model conditionally induces mutant KRAS in lung epithelium in *Nkx2-1* heterozygous mice upon doxycycline (Dox) administration (Maeda et al, 2012) while the other mouse model induces mutant KRAS and deletes *Nkx2-1* in mice carrying loxP alleles in lung cells by intratracheal adenovirus-expressing Cre recombinase (probably both epithelial cells as well as non-epithelial cells were infected by adenovirus; Snyder et al, 2013). Of note, NKX2-1 is expressed only in lung epithelial cells but not in non-epithelial lung cells. Actually, 113 genes were commonly induced in both the mouse models (25 genes + 88 genes; see Figs 1B and EV2); however only 25 genes among the 113 genes were also expressed in human IMA. Since this is the first attempt to create a gene signature for IMA, we did not want to miss any potential genes that molecularly represent IMA, so we included every human IMA gene that is expressed in either our mouse model or the other mouse model. With more specimens in the future, we may be able to reduce the number of the signature genes. In the revised manuscript, we sought to determine whether the 143 genes in the gene signature are expressed in the 9 datasets of human IMA from the 230 TCGA LUAD datasets. As stated in Materials and Methods, since the TCGA datasets do not have RNA-seq data for 8 out of 9 IMA-paired normal lung tissues, we randomly selected 9 datasets of normal lung tissues provided by the TCGA as controls. In addition, since the TCGA does not have the data for the expression of *MUC5AC* and *MUC3A/B*, we were able to assess whether the 141 genes from the 143 genes were expressed in the 9 datasets of human IMA from TCGA. As shown in Fig 1E, 140 genes out of the 141 genes were expressed in the 9 datasets of human IMA from the TCGA datasets, indicating that only one gene (*TNFSF18*) is inconsistent between our 6 IMA datasets and the 9 IMA datasets from the TCGA. We may decide to omit the one gene from the signature in the future when we obtain more datasets of IMA gene expression profiles. We discussed these issues in the discussion section in the revised manuscript.

Comment 3: In Figure 2E&G, the correlations between genes is quite different in the cell lines compare to the lung tumor tissue. The significance of this should be discussed

Response 3: In the revised manuscript, we reanalyzed the Pearson's correlations of gene expression in two datasets using the data transformations utilized in the original articles (Klijn et al, 2015; Cancer Genome Atlas Research Network, 2014), which are in log₂ scale and normalized with respect to library size. In the revised manuscript, Fig 3E (previously Fig 2E) and Fig 3G (previously Fig 2G) were re-generated using the updated correlation values. As pointed out, however, there is still a difference between the two datasets. The 230 TCGA LUAD dataset is from an intact lung tumor; thereby, it is composed of the combined gene expression profile of not only lung tumor cells but also tumor-associated fibroblasts, endothelial and immune cells, therefore it does not provide a gene expression profile of the solo lung tumor cells. The 105 NSCLC cell line dataset is composed of the gene expression profile of only tumor cells; however, each cell line is derived from an isolated tumor cell after multiple passages on plastic dishes, which may not represent lung tumor cells in an intact lung tumor environment. For example, genes such as *PD-L1* (*CD274/B7-H1*) and *VTCNI* (*B7-H4*) are expressed not only in lung tumor cells but also in immune cells. Thus, for *PD-L1* and *VTCNI*, the TCGA data cannot provide the gene-to-gene correlation only in lung tumor cells or immune cells, both of which compose the intact lung tumor environment. These issues lead to the differences in the correlation in the two datasets. We discussed the benefit and limitation of using the two datasets to assess the gene-to-gene correlation in the discussion section in the revised manuscript.

Comment 4: I am having a difficult time understanding Fig 2F. According to the legend this is hierarchical clustering of IMA-related genes and cell type markers. If the genes used are truly markers of IMA, how come they do such a poor job of clustering the IMA samples (indicated by red bars)?

Response 4: For the Fig 3D-G in the revised manuscript (Fig 2F in the previous manuscript), we used cell type markers for IMA as well as lung marker genes (e.g., surfactant proteins such as *SFTPB*). In the revised manuscript, we did the hierarchical clustering of the 230 TCGA LUAD cases using the gene signature of 141 genes (no *MUC5AC* and *MUC3A/B* expression in the 230 TCGA LUAD datasets). Convincingly using the gene signature, 5 of the human IMA cases from the TCGA datasets were highly clustered among the 230 TCGA LUAD cases (Fig 1D in the revised manuscript). Interestingly, the 5 cases carried *KRAS* mutations (Fig 1D and E), which suggests that our gene signature is superior in determining human IMA cases that carry *KRAS* mutations but not the ones that carry wild type *KRAS*, including fusion genes. We appreciate this comment, which led to the creation of new Fig 1D and E.

Comment 5: The data presented in Fig 3F &H does not seem that useful. I understand that the authors are trying to show that mucinous tumors rarely express PD-L1 but the percentage of non-mucinous tumors expressing PD-L1 is also quite low. So I am not sure how clinically relevant this information is. Although I am not an expert in this field, my understanding was that PD-L1 inhibitors have not really shown dramatic results in NSCLC patients.

Response 5: As reported previously (Herbst et al, Nature 2014), around 20% of non-small cell lung cancer (NSCLC) cases express PD-L1. As reviewed in a recent paper (Hirsch et al, Lancet 2016), lung cancer patients whose lung tumors express PD-L1 respond well to antibody-mediated therapies targeting PD-L1/PD-1, thus we consider that it is important to report that PD-L1 is expressed less in mucinous lung tumors (IMA) than in nonmucinous lung tumors (non-IMA). Blocking PD-L1 by therapeutic antibody has been shown to be clinically effective to treat NSCLC patients (Brahmer et al, NEJM 2012; Herbst et al, Nature 2014).

Some Minor Points

Comment 1: It would be useful if the authors used arrows or lines to clearly indicate which row belongs to each gene listed in Fig 1A.

Response 1: As suggested, we used lines to indicate which row belongs to each gene.

Comment 2: Stomach cancer and colorectal cancer cell enrichment scores have been reversed in the first line of page 7...colorectal cancer has enrichment score of 0.81 according to Fig 1E.

Response 2: We corrected that.

Referee #3 (Comments on Novelty/Model System):

The manuscript is well written. I have included some comments to the authors below that I hope will make the paper better. I find no reason to suspect ethical issues will result from this paper.

Referee #3 (Remarks):

The authors have attempted to elucidate the gene signature of Invasive Mucinous Adenocarcinoma (IMA), a rare type of lung adenocarcinoma (LUAD), and as of yet understudied. This signature turned up among other things, the potential gene target VTCN1. This manuscript appears well polished, well thought out and well written.

Major Critiques:

Comment 1: VTCN1 is proposed as a therapeutic target for IMA but is not found to be expressed at the protein level in all IMAs. How many conditions were tested for optimized VTCN1 staining? How many different clones of this antibody were tested? In the event it is only one, why? The reviewer would question, how do you propose to use this as a therapeutic target if it is not present? This leads me to question the robustness of the generated signature. Please discuss.

Response 1: As instructed by the manufacturer (Cell Signaling Technology, Danvers, MA), we used citrate for antigen retrieval (we used 1:100 dilution though 1:200 was recommended by the manufacturer). In order to validate this VTCN1 antibody (clone D1M8I; cat# 14572, Cell Signaling) that was used for IHC staining, we sought to independently determine by immunoblotting whether the antibody detects a specific band corresponding to VTCN1. First, we searched human lung cancer cell lines that express endogenous *VTCN1* and found that NCI-H1437 cells express endogenous VTCN1. Next, we knocked down VTCN1 in H1437 cells using three independent siRNAs and assessed the expression of VTCN1 at mRNA and protein levels. Three independent siRNAs significantly reduced the expression of *VTCN1* at the mRNA level. In addition, the three siRNAs reduced the protein bands detected by the VTCN1 antibody, indicating that this particular clone of the VTCN1 antibody is qualified to identify endogenous VTCN1. We added this data in the Appendix Fig S4 in the revised manuscript. Indeed, VTCN1 is expressed in 64% of the human IMA cases but not all, which is stated in the text. However, PD-L1 is not expressed in all of the NSCLC cases either. In fact, only around 20% of NSCLC cases express PD-L1 (Herbst et al, Nature 2014). Nonetheless, the 20% of the NSCLC patients whose lung tumors express PD-L1 have been shown to respond to antibody-mediated immunotherapy targeting PD-L1/PD-1 (Herbst et al, 2014; Garon et al, 2015; Hirsch et al, 2016). These papers are now cited in Results and Discussion in the revised text. It is ideal to identify immune checkpoints that express all of the lung cancer cases; however, currently, there is no such therapeutic target. Since around 80% of NSCLC patients do not express PD-L1 in their lung tumors, alternative targets for immune therapy are needed for NSCLC cases. In our present study, we found that 1) most human IMA cases do not express an immune checkpoint PD-L1; however, 2) a portion of human IMA cases express another immune checkpoint VTCN1. We believe that this portion of VTCN1-expressing human IMA patients can benefit from immunotherapy targeting VTCN1.

Comment 2: Related but in general, more important than catalogue numbers for your antibodies are the clone names. Please include. Catalogue numbers are of course useful, but to be truly reproducible, the same clone (for monoclonals) must be used between studies as well. Polyclonals are best avoided and their use appears limited in this manuscript.

Response 2: We included both catalogue numbers and clone names for all of the antibodies in Materials and Methods in the revised manuscript. The VTCN1 antibody is a monoclonal antibody.

Minor Critiques:

Comment 3: Data would be easier to interpret if the scales on 2D and 2F were the same; preferably 2D would be altered to fit scaling with 2F

Response 3: Fig 3D (Fig 2D in our previous manuscript) was re-generated using the variance-stabilizing transformation of gene expression data, which is the data transformation utilized in the original article (Klijn et al, 2015) for clustering analysis. Fig 3F (Fig 2F in our previous manuscript) was also re-generated using the normalized data downloaded from the original paper (Cancer Genome Atlas Research Network, 2014), which were quantified using RSEM (Li and Dewey, 2011), normalized within sample to a fixed upper quartile with zero values imputed by the overall minimum value and log2 transformed. The two data transformations are both in log2 scale, normalized with respect to library size and utilized in the original articles for clustering analysis. In Fig 3D and F in the revised manuscript, we further scaled the expression values using row zscore normalization.

Comment 4: Reference to Fig4G,H in the manuscript. There is no Fig4G and H. Do you mean Fig3G,H? Please revise. Reread your submissions carefully (including this one) to correct errors overlooked in manuscript writing and layout.

Response 4: We appreciated that the reviewer pointed out this error. We have carefully revised the manuscript and corrected other errors as well.

Minor Critiques:

Comment 5: Mucous production by tumors has been speculated to contribute to the invasive nature of these types of cancers. In order to foster ideas for the cancer research community branching out from this very rare cancer type, please discuss how this is relevant to your observations in pancreatic and breast tumors.

Response 5: MUC5AC and MUC5B are also highly expressed in pancreatic and breast cancers (Sóñora et al, 2006; Kaur et al, 2013). MUC5B is also reported to promote breast cancer growth (Valque et al, 2012). However, the mechanism by which these mucous genes are induced in pancreatic and breast cancers is not well understood. Analysis of the regulatory regions of these mucous genes, which were identified by our present study, in pancreatic and breast cancers will lead to the understanding of the gene regulatory network only in pancreatic and breast cancers but not in normal pancreatic and breast tissues. In the revised manuscript, we cited these papers and discussed this point in the discussion section.

Comment 6: IMA is known to be "defective in NKX2-1 expression" or lacking in NKX2-1 expression? Which is more accurate? Revise wording if necessary.

Response 6: Lacking in NKX2-1 expression is more accurate. We corrected that.

2nd Editorial Decision

04 November 2016

Thank you for the submission of your revised manuscript to EMBO Molecular Medicine. We have now heard back from the two Reviewers whom we asked to evaluate your manuscript.

You will see that, while reviewer 2 and 3 are satisfied that their concerns have been adequately addressed, reviewer 1 instead remains reserved and feels that a important issues remain pending that have not been adequately supported or discussed.

Specifically, reviewer 1 notes that the ChIP-seq data require a number of validation steps, which s/he outlines in detail. This reviewer also lists other concerns that require action.

After internal discussion, we agreed that the message of your study is compelling and deserves publication, but that the issues raised by reviewer 1 have merit and should be addressed. Although we would normally not allow a second extensive revision, based on our discussions I am prepared in this case however, to give you the opportunity to improve your manuscript by providing further experimentation and discussion of individual points as indicated by reviewer 1. The reviewer also notes your reference to "data not shown". S/he is correct that this is not permitted.

I remind you that EMBO Molecular Medicine has a "scooping protection" policy, whereby similar

findings that are published by others during review or revision are not a criterion for rejection. However, I do ask you to get in touch with us after three months if you have not completed your revision, to update us on the status. Please also contact us as soon as possible if similar work is published elsewhere.

I also suggest that you carefully adhere to our guidelines for publication in your next version, including our new requirements for supplemental data (see also below) to speed up the pre-acceptance process in case of a positive outcome. There are still a number of issues with the format that will need to be fixed, especially concerning the supplemental data. Do not hesitate to contact our editorial office for further assistance.

I look forward to seeing a revised form of your manuscript as soon as possible.

***** Reviewer's comments *****

Referee #1 (Remarks):

I recognize the time and effort invested by the authors revising their manuscript. The new version of the manuscript from Guo et al. contains additional experiments that further support the quality of the work and strength their interpretation of the data, thereby confirming my original opinion from the manuscript. Nevertheless, there is a concern that needs to be addressed before publication.

Concern:

As suggested in one of my major comments in the first round of review, all RNA-seq and ChIP-seq data were submitted to the Gene Expression Omnibus. In addition, a summary with relevant information from the NGS experiments is presented in the Table S8. Although the number of mapped reads (row I in Table S8) from the RNA-seq experiments is relatively low (4.8 to 7.7 millions for cell lines; 12.1 to 38.7 millions for human samples; see as reference Mortazavi et al 2008 PMID 18516045), I will not argue against the interpretation of the authors, since they have confirmed their RNA-seq experiments by TaqMan assays-based expression analysis presented in the Figure EV1. However, the ChIP-seq experiments were not confirmed by an alternative method although they also had low number of mapped reads (5 to 18 millions; see as reference Sims et al 2014 PMID 24434847 and Landt et al 2012 PMID 22955991). Even though the author presented replicates for each of the ChIP-seq experiments, the sequencing depth is a critical aspect in the NGS experiments as stated in the references mentioned above.

In addition, even though the authors presented reduced expression of MUC5AC (Figure 7D) and MUC5B (Figure 8D) after depleting the region that was identified in their ChIP-seq experiments to be bound by SPDEF using CRISPR/Cas9 technology, they cannot conclude that MUC5AC and MUC5B are direct targets from SPDEF. The binding elements from other transcription factors were also depleted, thereby questioning the causality of the effects observed. Directed mutations to the SPDEF binding element by CRISPR/Cas9 technology would have been a cleaner experiment. Due to the time and effort related to this type of experiments, I will not request the authors to perform them. NEVERTHELESS, I would like to suggest the following experiments in order of relevance to confirm the ChIP-seq results:

A. ChIP based analysis of MUC5AC and MUC5B after immunoprecipitating endogenous SPDEF.

B. If the authors have concerns with the low amount of SPDEF in the cells, or with the quality of the antibody for ChIP experiments, I would like to suggest an overexpression of Myc-SPDEF or Flag-SPDEF and subsequent ChIP analysis of MUC5AC and MUC5B after immunoprecipitating overexpressed SPDEF using a commercial available antibodies against the Myc- or Flag-tag.

C. Overexpression of SPDEF in the cells used in the Figure 7D should increase the mRNA levels of MUC5B without affecting the levels of MUC5AC. A similar experiment could be performed for the cells used in the Figure 8D.

The results obtained from the experiments A OR B will allow the authors to demonstrate that

MUC5AC and MUC5B are direct targets from SPDEF. In addition, the results from the experiment C will confirm that the reduction in the expression of these genes observed after the depletions induced by CRISPR/Cas9 technology in the Figures 7D and 8D are caused by a lack of binding of SPDEF. In other words, experiment C will confirm the causality of SPDEF on the effects observed after CRISPR/Cas9-mediated depletions.

Minor concerns:

- The fact that SPDEF or FOXA3 in combination with mutant KRAS induce malign or benign mucinous lung tumors respectively, highlighting mechanistic differences during tumorigenesis, is important and should be mentioned in the abstract.
- The last part of the Introduction and the initial part of the Discussion are redundant, since they repeat the results, which in any case are also described in the corresponding Result section. The authors should avoid redundancy.
- The expression analysis presented in the Figure S2 is relevant to the conclusion of the authors and should be presented in the main Figure 6, perhaps after Figure 6A.
- In Response 4 of the rebuttal letter, the authors commented on the fact that although a reduction of NKX2-1 in IMA at the protein level is well established by immunohistochemistry (Travis et al, 2011); the mRNA levels of NKX2-1 were not significantly reduced in IMA when compared to normal lung tissue in the RNA-seq experiment (Fig 1A) and the Taqman gene expression qPCR analyses (Fig EV1; P=0.051).

Interestingly, Mehta et al., 2016 (DOI 10.15252/emmm.201606382) also detected expression of NKX2-1 in all three subtypes of NSCLC. A plausible explanation for these observations might be that transcript detection is more sensitive than immunodetection. Supporting this line of ideas, amplification of NKX2-1 has been reported in 20% of 99 SQCC cases analyzed by FISH but not detected at the protein level (Tang et al, 2011) and 14q13.3 amplification, containing NKX2-1, is one of the most significant amplifications in SQCC reported in TCGA.

The authors should mention these observations in the discussion, because this will help to improve the use of NKX2-1 as marker for different lung cancer subtypes. One good place to mention these observations might be after their statements:

"...Our present data shows that NKX2-1, which is absent in IMA (Travis et al, 2011), induced PD-L1 in mucinous lung cancer cells in vitro (Fig 4), suggesting that the lack of PD-L1 in IMA is due to the absence of NKX2-1. NKX2-1 did not induce PD-L1 nor did the inhibition of NKX2-1 by siRNA reduce PD-L1 in non-mucinous lung cancer cells (data not shown), suggesting that induction of PD-L1 by NKX2-1 is unique to mucinous lung cancer cells."

- In the current Figure 6B, please change the color combination red-orange, to other color combination in order that the two different groups can be better recognized.
- The authors mention in the Discussion section own data as data not shown. One has to check the guide lines from EMM, whether this is allowed.

I hope that my suggestion contribute to further improve the quality of the manuscript to reach the standard for publication at EMM.

Referee #2 (Remarks):

The author has addressed all of my concerns

Referee #3 (Comments on Novelty/Model System):

The publication will provide useful information for translational therapeutics against IMA.

Referee #3 (Remarks):

The authors have sufficiently answered all of my questions and corrected errors within the manuscript.

2nd Revision - authors' response

08 January 2017

Referee #1 (Remarks):

I recognize the time and effort invested by the authors revising their manuscript. The new version of the manuscript from Guo et al. contains additional experiments that further support the quality of the work and strength their interpretation of the data, thereby confirming my original opinion from the manuscript. Nevertheless, there is a concern that needs to be addressed before publication.

Concern:

Comment 1: As suggested in one of my major comments in the first round of review, all RNA-seq and ChIP-seq data were submitted to the Gene Expression Omnibus. In addition, a summary with relevant information from the NGS experiments is presented in the Table S8. Although the number of mapped reads (row I in Table S8) from the RNA-seq experiments is relatively low (4.8 to 7.7 millions for cell lines; 12.1 to 38.7 millions for human samples; see as reference Mortazavi et al 2008 PMID 18516045), I will not argue against the interpretation of the authors, since they have confirmed their RNA-seq experiments by TaqMan assays-based expression analysis presented in the Figure EV1. However, the ChIP-seq experiments were not confirmed by an alternative method although they also had low number of mapped reads (5 to 18 millions; see as reference Sims et al 2014 PMID 24434847 and Landt et al 2012 PMID 22955991). Even though the author presented replicates for each of the ChIP-seq experiments, the sequencing depth is a critical aspect in the NGS experiments as stated in the references mentioned above.

In addition, even though the authors presented reduced expression of MUC5AC (Figure 7D) and MUC5B (Figure 8D) after depleting the region that was identified in their ChIP-seq experiments to be bound by SPDEF using CRISPR/Cas9 technology, they cannot conclude that MUC5AC and MUC5B are direct targets from SPDEF. The binding elements from other transcription factors were also depleted, thereby questioning the causality of the effects observed. Directed mutations to the SPDEF binding element by CRISPR/Cas9 technology would have been a cleaner experiment. Due to the time and effort related to this type of experiments, I will not request the authors to perform them. NEVERTHELESS, I would like to suggest the following experiments in order of relevance to confirm the ChIP-seq results:

A. ChIP based analysis of MUC5AC and MUC5B after immunoprecipitating endogenous SPDEF.

B. If the authors have concerns with the low amount of SPDEF in the cells, or with the quality of the antibody for ChIP experiments, I would like to suggest an overexpression of Myc-SPDEF or Flag-SPDEF and subsequent ChIP analysis of MUC5AC and MUC5B after immunoprecipitating overexpressed SPDEF using a commercial available antibodies against the Myc-or Flag-tag.

C. Overexpression of SPDEF in the cells used in the Figure 7D should increase the mRNA levels of MUC5B without affecting the levels of MUC5AC. A similar experiment could be performed for the cells used in the Figure 8D.

The results obtained from the experiments A OR B will allow the authors to demonstrate that MUC5AC and MUC5B are direct targets from SPDEF. In addition, the results from the experiment C will confirm that the reduction in the expression of these genes observed after the depletions induced by CRISPR/Cas9 technology in the Figures 7D and 8D are caused by a lack of binding of SPDEF. In other words, experiment C will confirm the causality of SPDEF on the effects observed after CRISPR/Cas9-mediated depletions.

Response 1: As to the suggested experiment A or B, we performed ChIP-qPCR using the SPDEF antibody and confirmed that SPDEF bound to the enhancers of MUC5AC and MUC5B in A549 lung

carcinoma cells that express ectopic SPDEF, which is now shown in Fig EV5E in the revised manuscript. The SPDEF antibody (cat# sc-67022X; Santa Cruz Biotechnology) that we used in our study was previously used by Dr. Kerstin Meyer's group at the University of Cambridge to identify SPDEF binding sites by ChIP-seq in MCF7 breast adenocarcinoma cells (Fletcher et al, Nat Commun 2013). In the revised manuscript, we stated in Material and Methods that the antibody was previously used for ChIP-seq by Fletcher et al. As we have shown in our previous manuscripts, in order to validate our ChIP-seq data, we downloaded Fletcher et al's ChIP-seq data and loaded their bed files along with our ChIP-seq bed files on the loci of *MUC5AC* and *MUC5B*, which are shown in Figs 7A and 8A. Importantly, our duplicated ChIP-seq peaks on the enhancer region of *MUC5AC* were aligned with one of the Fletcher et al's triplicated ChIP-seq peaks (Fig 7A; please note that MCF7_SPDEF_peaks rep1-3 are from Fletcher et al). Further importantly, our duplicated ChIP-seq peaks on the enhancer region of *MUC5B* were aligned with all three of the Fletcher et al's triplicated ChIP-seq peaks (Fig 8A; please note that MCF7_SPDEF_peaks rep1-3 are from Fletcher et al). These results indicate that two independent groups (Dr. Meyer's group and our group) reached the same conclusion that SPDEF directly associates with the loci of *MUC5AC* and *MUC5B*. In order to further validate our ChIP-seq data, we performed the motif enrichment analysis recommended by Landt et al, Genome Res 2012, which reviewer 1 referenced. The motif enrichment analysis by Dr. Meyer's group and our group using their and our ChIP-seq datasets led to the identification of the SPDEF binding motif (Fig EV5 in our manuscript; Fletcher et al, Nat Commun 2013). Thus, we believe that our ChIP-seq data has been sufficiently validated.

Figure. Three (A, B or C) potential regulations of mucin genes by SPDEF through SPDEF binding sites

Regarding the experiment C suggested by reviewer 1, *MUC5AC* and *MUC5B* are located in the same chromosome 11 locus harboring four mucin genes *MUC6*, *MUC2*, *MUC5AC* and *MUC5B* (see Figure above). There are at least three SPDEF binding sites, including the two enhancers of *MUC5AC* and *MUC5B*. After we deleted the enhancer of *MUC5AC* or *MUC5B* by CRISPR/Cas9,

the expression of *MUC5AC* or *MUC5B* mRNA was reduced compared to the control; however, expression was not completely diminished. There is a possibility that the residual expression of *MUC5AC* and *MUC5B* mRNA might be regulated and/or induced by SPDEF through other SPDEF binding sites that were not deleted. Such regulation by enhancer regions has been recently reviewed by Dr. Richard Young and colleagues (Hnisz et al, Cell 2016). In order to elucidate the mechanism by which enhancers are indispensable and/or dispensable for SPDEF to induce *MUC5AC* and *MUC5B*, we would need to delete all of the three enhancers simultaneously. However, it would not be feasible to produce such a cell line in a short time frame. As we cited in the Discussion section in our manuscript, the same approach was taken by Drs. Richard Young and A. Thomas Look's group at MIT and Dana-Farber Cancer Institute to look at the endogenous expression of *TALI* in Jurkat cells when its enhancer region is deleted by CRISPR/Cas9 (Mansour et al, Science 2014). Dr. Matthew Meyerson's group at Dana-Farber Cancer Institute also took the same approach to look at the endogenous expression of *MYC* in H2009 cells when its enhancer region is deleted by CRISPR/Cas9 (Zhang et al, Nat Genet 2016). Both studies deleted only one enhancer region but not multiple regions in an endogenous context. We agree with reviewer 1 that further mechanistic studies would be ideal beyond previously published studies, which would take more time; however, we believe that we currently have enough data and that publication at this time would be useful for lung researchers and potentially benefit lung cancer patients.

Minor concerns:

Comment 2: The fact that SPDEF or FOXA3 in combination with mutant KRAS induce malign or benign mucinous lung tumors respectively, highlighting mechanistic differences during tumorigenesis, is important and should be mentioned in the abstract.

Response 2: As instructed, we stated "benign or malignant" in the abstract in the revised manuscript. We were not able to state further pathological differences in the abstract due to the word limitation; however, the details are stated in the Results.

Comment 3: The last part of the Introduction and the initial part of the Discussion are redundant, since they repeat the results, which in any case are also described in the corresponding Result section. The authors should avoid redundancy.

Response 3: Since the two other reviewers did not raise this issue and the third reviewer commented that the manuscript is well written, we are afraid to change the readiness of the current text by the editing suggested. We would be happy to change as suggested if the editor instructs so.

Comment 4: The expression analysis presented in the Figure S2 is relevant to the conclusion of the authors and should be presented in the main Figure 6, perhaps after Figure 6A.

Response 4: As suggested by the reviewer, the data presented in Appendix Fig S2 in the previous manuscript is now included in Fig 6 in the revised manuscript.

Comment 5: In Response 4 of the rebuttal letter, the authors commented on the fact that although a reduction of NKX2-1 in IMA at the protein level is well established by immunohistochemistry (Travis et al, 2011); the mRNA levels of NKX2-1 were not significantly reduced in IMA when compared to normal lung tissue in the RNA-seq experiment (Fig 1A) and the Taqman gene expression qPCR analyses (Fig EV1; P=0.051).

Interestingly, Mehta et al., 2016 (DOI 10.15252/emmm.201606382) also detected expression of NKX2-1 in all three subtypes of NSCLC. A plausible explanation for these observations might be that transcript detection is more sensitive than immunodetection. Supporting this line of ideas, amplification of NKX2-1 has been reported in 20% of 99 SQCC cases analyzed by FISH but not detected at the protein level (Tang et al, 2011) and 14q13.3 amplification, containing NKX2-1, is one of the most significant amplifications in SQCC reported in TCGA.

The authors should mention these observations in the discussion, because this will help to improve the use of NKX2-1 as marker for different lung cancer subtypes. One good place to mention these observations might be after their statements: "...Our present data shows that NKX2-1, which is absent in IMA (Travis et al, 2011), induced PD-L1 in mucinous lung cancer cells in vitro (Fig 4), suggesting that the lack of PD-L1 in IMA is due to the absence of NKX2-1. NKX2-1 did not induce PD-L1 nor did the inhibition of NKX2-1 by siRNA reduce PD-L1 in non-mucinous lung cancer cells (data not shown), suggesting that induction of PD-L1 by NKX2-1 is unique to mucinous lung cancer cells."

Response 5: It is our understanding in reading comment 5 that reviewer 1 appears to consider transcript detection to be more sensitive than immunodetection based on the Tang et al, 2011 paper. However, the Tang et al, 2011 paper investigated DNA amplification (copy number gain) of *NKX2-1* (also known as *TTF-1* or *TTF-1*) but not RNA transcript level of *NKX2-1*. Thus, the Tang et al, 2011 paper does not support the assumption or concept that transcript detection is more sensitive than immunodetection. Regarding the immunodetection of NKX2-1, the reduction of NKX2-1 protein in IMA compared to normal lung tissue is an observation made in clinical specimens, which was established by an international core panel of experts representing the International Association for the Study of Lung Cancer, American Thoracic Society, and European Respiratory Society (Travis et al, J Thorac Oncol 2011). In our hands, we also observed the reduction of NKX2-1 protein in IMA compared to normal lung tissue using immunohistochemistry (Fig 4E). Thus, our data is consistent with the observation by Travis et al 2011. Regarding the expression of *NKX2-1* mRNA, we also observed reduction of *NKX2-1* levels were ~50% reduced in IMA compared to normal lung tissues (Fig EV1); however, this difference did not reach statistical significance ($P=0.051$) based on the number of samples available for our study (Fig EV1).

Comment 6: In the current Figure 6B, please change the color combination red-orange, to other color combination in order that the two different groups can be better recognized.

Response 6: We changed the color from orange to green.

Comment 7: The authors mention in the Discussion section own data as data not shown. One has to check the guide lines from EMM, whether this is allowed.

Response 7: We removed the sentence from the text.

Comment 8: I hope that my suggestion contribute to further improve the quality of the manuscript to reach the standard for publication at EMM.

Response 8: We hope that we improved the quality of our manuscript and that it now reaches the standard for publication in EMM.

Referee #2 (Remarks): The author has addressed all of my concerns

*Referee #3 (Comments on Novelty/Model System):
The publication will provide useful information for translational therapeutics against IMA.*

Referee #3 (Remarks): The authors have sufficiently answered all of my questions and corrected errors within the manuscript.

2nd Editorial Decision

20 January 2017

Thank you for the submission of your revised manuscript to EMBO Molecular Medicine. We have now received the enclosed report from the reviewer who was asked to re-assess it. As you will see s/he are now supportive and I am pleased to inform you that we will be able to accept your manuscript pending the following final editorial amendments.

Please submit your revised manuscript within two weeks. I look forward to seeing a revised form of your manuscript as soon as possible.

***** Reviewer's comments *****

Referee #1 (Remarks):

The authors have sufficiently addressed our concerns during peer review. Regarding one of the major observations, they included additional ChIP-qPCR analysis to validate their ChIP-seq data. Importantly, they cross validated their ChIP-seq results (duplicates) with available published data (Fletcher et. al., Nat Commun 2013, triplicates) to state their major conclusion about SPDEF-dependent occupancy at the MUC5-AC/B loci. Both analyses strengthen their observations despite the low number of mapped reads.

Minor concerns were either sufficiently answered or justified. In the last case, I assume that in science one can have different opinions in certain points...

Summarizing, I re-confirmed my opinion about the quality and the novelty of the work. I would like to recommend the manuscript for publication at EMBO Mol Med.

3rd Revision - authors' response

24 January 2017

Authors made requested editorial changes.

Corresponding Author Name: Yutaka Maeda

Manuscript Number: EMM-2016-06711